# Inversion by Direct Iteration:
# An Alternative to Denoising Diffusion for Image Restoration

**Mauricio Delbracio**                                    *mdelbra@google.com*
*Google Research*

**Peyman Milanfar**                                    *milanfar@google.com*
*Google Research*

**Reviewed on OpenReview:** *https://openreview.net/forum?id=VmyFF5lL3F*

## Abstract

Inversion by Direct Iteration (InDI) is a new formulation for supervised image restoration that avoids the so-called "regression to the mean" effect and produces more realistic and detailed images than existing regression-based methods. It does this by gradually improving image quality in small steps, similar to generative denoising diffusion models.

Image restoration is an ill-posed problem where multiple high-quality images are plausible reconstructions of a given low-quality input. Therefore, the outcome of a single step regression model is typically an aggregate of all possible explanations, therefore lacking details and realism. The main advantage of InDI is that it does not try to predict the clean target image in a single step but instead gradually improves the image in small steps, resulting in better perceptual quality.

While generative denoising diffusion models also work in small steps, our formulation is distinct in that it does not require knowledge of any analytic form of the degradation process. Instead, we directly learn an iterative restoration process from low-quality and high-quality paired examples. InDI can be applied to virtually any image degradation, given paired training data. In conditional denoising diffusion image restoration the denoising network generates the restored image by repeatedly denoising an initial image of pure noise, conditioned on the degraded input. Contrary to conditional denoising formulations, InDI directly proceeds by iteratively restoring the input low-quality image, producing high-quality results on a variety of image restoration tasks, including motion and out-of-focus deblurring, super-resolution, compression artifact removal, and denoising.

## 1  Introduction

Recovering a high-quality image from a low-quality observation is a fundamental problem in computer vision and computational imaging. Single image restoration is a highly ill-posed inverse problem where multiple plausible sharp and clean images could lead to the very same degraded observation. The typical supervised approach is to formulate image restoration as a problem of inferring the underlying image given a low-quality version of it, by training a model with paired examples of the relevant degradation (Ongie et al., 2020). One of the most common approaches is to directly minimize a pixel reconstruction error using the $L_1$ or $L_2$ loss; an approach that correlates well with the popular PSNR (peak signal-to-noise-ratio) metric. However, it has been observed often in recent literature that measures such as PSNR (and in general point-distortion metrics) do not correlate well to human perception (Blau & Michaeli, 2018; Delbracio et al., 2021b; Freirich et al., 2021). Despite these shortcomings, much of the recent research work has been focused on improving deep architectures and optimizing a variety of point-loss formulations, resulting in general models that give an aggregate improved image in one step of inference.

To see the issues more concretely, let's assume that we are given image pairs $(\boldsymbol{x}, \boldsymbol{y}) \sim p(\boldsymbol{x}, \boldsymbol{y})$ where $\boldsymbol{x}$ represents a target high-quality image, and $\boldsymbol{y}$ represents the respective degraded observation. For instance, $\boldsymbol{x}$ may be pristine images degraded by a combination of blur/compression and noise to yield $\boldsymbol{y}$. A typical regression approach would predict $\boldsymbol{x}$ directly from $\boldsymbol{y}$ using a trained model $\hat{\boldsymbol{x}}(\boldsymbol{y}) = F_\theta(\boldsymbol{y}) \approx \boldsymbol{x}$, by minimizing the expected pixel error in some (e.g. $L_p$) metric as follows:

$$\min_\theta \mathbb{E}_{\boldsymbol{x}, \boldsymbol{y}} \|F_\theta(\boldsymbol{y}) - \boldsymbol{x}\|_p \approx \min_\theta \sum_i \|F_\theta(\boldsymbol{y}^i) - \boldsymbol{x}^i\|_p.$$

In the case $p = 2$, the minimum mean-squared error (MMSE) optimal solution is the conditional expectation: $\boldsymbol{x}_{\text{MMSE}}(\boldsymbol{y}) = \mathbb{E}[\boldsymbol{x} \,|\, \boldsymbol{y}] = \int \boldsymbol{x} p(\boldsymbol{x} \,|\, \boldsymbol{y}) d\boldsymbol{x}$.

This evidently results in an image that is the (weighted) average of all plausible reconstructions[1]. This resulting image will not have a natural appearance as the details have been wiped out due to the effect aggregation (i.e. "regression to the mean") effect. The problem is compounded for more ill-posed problems. That is, the more ill-posed the inverse problem, the larger the set of plausible reconstructions and therefore the more severe the effect of the aggregation implied by the expectation of the posterior. To mitigate this problem, recent works have introduced additional loss terms (Gatys et al., 2016; Mechrez et al., 2018b;a; Delbracio et al., 2021b; Kupyn et al., 2018) that seek a balance in the formulation so the final image has improved perceptual quality (more on this in the next section).

In this work, we explicitly address this problem by avoiding single-step prediction of the clean image, and instead iterating a series of inferences, where at each step we solve an 'easier' (i.e., less ill-posed) inverse problem than the original. Specifically, we generate a sequence of intermediate restorations where at each step the goal is to reconstruct only a slightly less corrupted image. The core observation underlying this approach is this: *a small-step restoration largely avoids the regression-to-the-mean effect because the set of plausible 'slightly-less-bad' images is relatively small.* The core technical component enabling our approach is still a single deep model, but one that is trained to predict a better image given one with an intermediate level of degradation in the previous step, as summarized by Algorithm 1.

## 2  Background

Recently, much work on imaging inverse problems has been focused on using generative formulations (Bora et al., 2017; Kawar et al., 2022). Generative adversarial formulations train restoration networks with an adversarial loss that forces the restored image to be on the distribution of high-quality signals (Kupyn et al., 2018; 2019; Asim et al., 2020). GANs are in general hard to train and also hard to control image hallucinations since the two terms play an antagonic role (Lugmayr et al., 2021).

Image priors, including generative ones, can be used to solve inverse problems in an unsupervised fashion where the degradation operator is only know at inference time (Rudin & Osher, 1994; Venkatakrishnan et al., 2013; Delbracio et al., 2021a; Romano et al., 2017; Ongie et al., 2020). DDPMs have been recently adapted for unsupervised model-based image restoration (Kawar et al., 2021a; 2022; Kadkhodaie & Simoncelli, 2021; Jalal et al., 2021a; Laumont et al., 2022; Chung et al., 2022; Kawar et al., 2022).

**How this approach compares to Denoising Diffusion.** Denoising Diffusion Probabilistic Models (DDPMs) (Sohl-Dickstein et al., 2015; Ho et al., 2020; Song et al., 2021a) and Score-based models (Song & Ermon, 2019; 2020; Song et al., 2021b) have emerged as two powerful classes of generative models that produce high-quality samples by inverting a *known* diffusion (degradation) process. The standard Gaussian denoising formulation has been extended to more general corruption process (Bansal et al., 2022; Hoogeboom & Salimans, 2022; Daras et al., 2023; Deasy et al., 2021; Hoogeboom et al., 2022a;b; Nachmani et al., 2021; Johnson et al., 2021; Lee et al., 2022; Ye et al., 2022). The main common idea is to analytically define a known degradation process that is reversed to generate new samples starting from a fully degraded image (e.g., pure noise). The inference procedure makes use of the known analytical degradation at every step.

---

[1] A similar statement is true for other $p \neq 2$ in which case the mean is replaced by another aggregation operator (e.g. median for $L_1$)

By contrast, in our formulation we do not require knowledge of any analytic form of the degradation process, we directly learn an iterative restoration process from low-quality/high-quality paired examples. This implies that we can apply our iterative procedure to virtually any degradation as long as we are given image pairs. Additionally, our formulation is motivated only from the idea of splitting the original inverse problem into multiple smaller ones. We do not require any knowledge of the underlying probability distributions, or the conditional distributions, at any step. Our inference procedure is solely based on the idea of restoring the signal a little bit at each step. This approach, with minimal assumptions, gives a unified formulation for any supervised image restoration problem under the same framework.

The natural extension of DDPMs to image restoration tasks is through the use of a *Conditional* DDPM models (cDDPM) (Li et al., 2021; Saharia et al., 2021; 2022; Whang et al., 2022). The goal of a cDDPM is to generate plausible reconstructions given the low-quality input (e.g., by generating samples from the posterior distribution). The idea is to train a supervised *denoising* diffusion model using paired examples that is conditioned on the low-quality input. The denoising network learns to generate a valid restored image (sample) by repeatedly denoising an initial image of pure noise. Our formulation has some similarity to conditional diffusion models, but contrary to the denoising formulation, we directly proceed by iteratively restoring the input image.

Overall, our method is straightforward to implement and train, and produces high-quality results. We evaluate the formulation on four different restoration tasks using different perceptual quality metrics. As shown, our method produces samples of higher quality than the state-of-the-art regression formulations while maintaining high-fidelity with respect to the original sample.

## 3 Related Work

The goal of image restoration is to generate a high-quality image from its degraded low-quality measurement (e.g., low-resolution, compressed, noisy, blurry). Since the seminal super-resolution work of Dong et al. (2015) many recent image restoration methods adopt an end-to-end supervised formulation where a deep neural network is trained to directly produce a point estimate (Zhao et al., 2016; Lim et al., 2017; Tao et al., 2018; Chen et al., 2018) These methods rely on low-quality high-quality image pairs to train a regression model. Most of the work has been focused on developing better and more powerful network architectures (Zamir et al., 2022; Chen et al., 2022; Tu et al., 2022; Zamir et al., 2021) so we can achieve better pixel-level reconstruction. While this formulation leads to state-of-the-art PSNR, the image generated is at best an average of all plausible solutions (regression to the mean). In the limit case where the low-quality image is completely obfuscated the best prediction in terms of PSNR is the average of the distribution.

Generative adversarial networks (Goodfellow et al., 2014; Arjovsky et al., 2017), and adversarial formulations (Ledig et al., 2017; Isola et al., 2017; Kupyn et al., 2018; 2019) have been introduced to push the generated image towards the manifold of natural images. GANs suffer from unstable training (Arora et al., 2017; Salimans et al., 2016; Arjovsky et al., 2017), while being prone to introduce significant image hallucinations. This is a direct consequence of a non-reference formulation that directly tries to minimize the distance of the range of the generator to the manifold of natural images (Cohen et al., 2018).

Blau & Michaeli (2018) proved that there is a trade-off between image perceptual quality and distortion. It is not possible to minimize both distortion and perceptual quality simultaneously. In fact, minimizing the average point distortion (e.g., PSNR) can be only done in detriment of the perceptual quality (Blau & Michaeli, 2018; Freirich et al., 2021).

A powerful way of avoiding the regression to the mean is to formulate the problem as one of sampling from the posterior distribution (Kawar et al., 2021b;a; 2022; Ohayon et al., 2021; Kadkhodaie & Simoncelli, 2021; Whang et al., 2022). An additional benefit of this formulation is to be able to generate multiple different plausible solutions that can be used for uncertainty quantification (Whang et al., 2021) or improving fairness (Jalal et al., 2021b).

Variational auto-encoders (Prakash et al., 2020), Normalizing flows (Lugmayr et al., 2020; 2021), and Diffusion probabilistic models (DPMs) (Saharia et al., 2021; Li et al., 2021; Whang et al., 2022) have been

successfully applied to different image restoration tasks, where a diverse set of candidates can be generated from the learned posterior (Prakash et al., 2020).

Denoising Diffusion Probabilistic Models (DDPMs) (Sohl-Dickstein et al., 2015; Ho et al., 2020; Song et al., 2021a), Score-based models (Song & Ermon, 2019; 2020; Song et al., 2021b) and their recent generalizations (Bansal et al., 2022; Hoogeboom & Salimans, 2022; Daras et al., 2023; Deasy et al., 2021; Hoogeboom et al., 2022a;b; Nachmani et al., 2021; Johnson et al., 2021; Lee et al., 2022; Ye et al., 2022) generate high-quality samples by inverting a known degradation process. The main strategy is to analytically define a known degradation process that is reversed to generate new samples starting from a fully degraded image (e.g., pure noise). Bansal et al. (2022) introduced Cold Diffusion a generative framework to generate images by reverting arbitrary (known) degradations. They show promising results even with non-stochastic degradations such as blur, masking or pixelization. The strategy is to define intermediate analytical degradations (diffusion) and then revert them step by step. Daras et al. (2023) presented Soft Diffusion a generalization of diffusion models to linear degradations. The authors argue that noise is a fundamental component that is needed to provably learn the score.

In this work, we adopt a similar strategy and propose to decompose the image restoration problem into a sequence of intermediate steps each of them being a much easier problem to solve (i.e., less ill-posed). This path of intermediate reconstructions takes us from a low-quality input to a high-quality reconstruction through a series of slightly less corrupted signals. Different than in traditional generative diffusion formulations, the degradation is only *implicitly* given through a series of pair images (low-quality and high-quality). In our formulation, we define the intermediate steps as a convex combination of the target/input signals. This induces a simple linear propagation from the high-quality sample to the low-quality one.

An alternative formulation of the supervised image restoration problem is to use a conditional denoising diffusion models to generate samples from the posterior distribution (Li et al., 2021; Saharia et al., 2021; 2022; Whang et al., 2022). The overall idea is to train a denoising diffusion model that is conditioned on the low-quality input. The denoising network learns to generate a valid restored image by repeatedly denoising an initial image of pure noise. Our formulation has some similarity to conditional diffusion models, but contrary to the denoising formulation, we directly proceed by iteratively restoring the input image.

The very recent work by Luo et al. (2023a) and Welker et al. (2022), and the concurrent work by Luo et al. (2023b); Song et al. (2023) introduce related image restoration techniques based on ODE/SDE diffusion formulations. A major difference with our work is that InDI is completely motivated and formulated from elementary principles by splitting the restoration tasks into multiple smaller ones. There is also significant amount of recent related work that analyzes the connection of diffusion image generation with Bridge and Flow matching, Optimal Transport and Schrodinger bridges (Albergo et al., 2023; Shi et al., 2023; Liu et al., 2023). Finally, the concurrent work of Heitz et al. (2023) adopts a linear diffusion scheme for image generation similar to, but less general than, the one in InDI.

## 4 InDI: Our Proposed Formulation

Given $(\boldsymbol{x}, \boldsymbol{y}) \sim p(\boldsymbol{x}, \boldsymbol{y})$, we define a continuous forward degradation process by

$$\boldsymbol{x}_t = (1 - t)\boldsymbol{x} + t\boldsymbol{y}, \quad \text{with } t \in [0, 1]. \tag{1}$$

The idea of this forward process is that it starts from a clean sharp image at time $t = 0$, and then degrades it to the blurry/noisy observation at time $t = 1$. Here, $\boldsymbol{x}_t$ indexed by $t$, represents an intermediate degraded image between the low-quality input $\boldsymbol{y}$ (i.e., $t = 1$) and the high-quality sharp target $\boldsymbol{x}$ (i.e., $t = 0$). Following the common notation in diffusion models, we will refer to the index $t$ as the time-step.

Our recovery method starts with the input degraded image (time $t = 1$), and then at a given time-step $t$ generates the best possible reconstruction at time $t - \delta$. This can be done, for example, by the short-time conditional mean $\hat{\boldsymbol{x}}_{t-\delta} = \mathbb{E}[\boldsymbol{x}_{t-\delta} \mid \hat{\boldsymbol{x}}_t]$ to the estimate. As we will show, by repeating this process we can thus invert the full degradation little by little. The following proposition provides the cornerstone of our approach. f

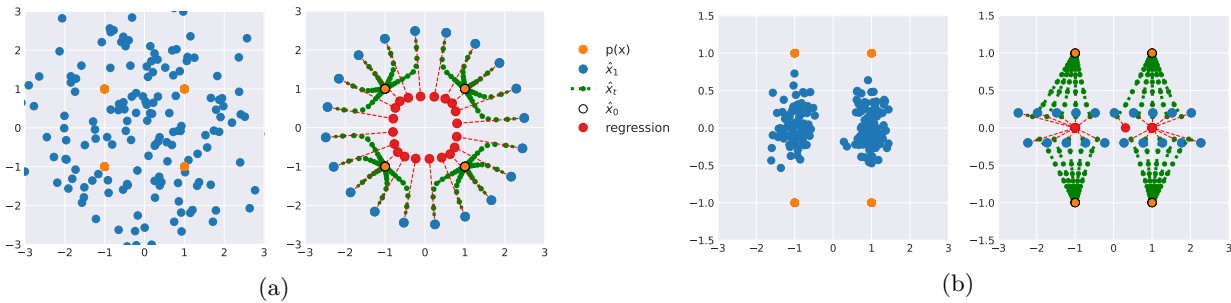

Figure 1: 2D Toy Example. Estimation of conditional mean and iterated estimation for points from a multimodal (4 modes) distribution under: (a) Denoising strong Gaussian noise ($\boldsymbol{H} = \boldsymbol{I}$); and (b) missing information recovery, i.e., $\boldsymbol{H} = [1,0;0,0]$, under moderate noise. Blue points represent observed samples, while red ones are the *regression* prediction. The black (hollow) circles represent the final point in our iterative procedure, always reaching a valid point in the data manifold (orange points). The small green circles indicate the iterative restoration path.

**Proposition 4.1.** *Let $\boldsymbol{x}_s, \boldsymbol{x}_t$ be given from equation 1, where $s \leq t$. Then,*

$$\mathbb{E}[\,\boldsymbol{x}_s\,|\,\boldsymbol{x}_t\,] = \left(1 - \frac{s}{t}\right)\mathbb{E}[\,\boldsymbol{x}_0\,|\,\boldsymbol{x}_t\,] + \frac{s}{t}\boldsymbol{x}_t.$$

The proof is a direct consequence of change of variables and is given in Appendix A.

According to this proposition, the posterior mean (e.g. MMSE estimate) at time $s < t$ can be deduced from the estimate at time $t$ by first estimating the clean image ($\boldsymbol{x} = \boldsymbol{x}_0$), and then doing a convex combination with the estimate $\hat{\boldsymbol{x}}_s$ at time $s$. We can then apply the following scheme to move from $t$ to $s = t - \delta$,

$$\hat{\boldsymbol{x}}_{t-\delta} = \mathbb{E}[\,\boldsymbol{x}_{t-\delta}\,|\,\hat{\boldsymbol{x}}_t\,] = \frac{\delta}{t}\mathbb{E}[\,\boldsymbol{x}_0\,|\,\hat{\boldsymbol{x}}_t\,] + \left(1 - \frac{\delta}{t}\right)\hat{\boldsymbol{x}}_t. \tag{2}$$

The process starts from $\hat{\boldsymbol{x}}_1 = \boldsymbol{y}$, and the step $\delta < 1$ controls the "speed" of the reverse process (e.g., at constant "speed", $\delta = \frac{1}{N}$, where $N$ controls the total number of steps).

**Remark 4.2.** *For the iteration procedure in equation 2 to be well defined we require that $\mathbb{E}[\,\boldsymbol{x}_0\,|\,\hat{\boldsymbol{x}}_t\,]$ is well defined. Thus, we require $p_{\boldsymbol{x}_t}(\hat{\boldsymbol{x}}_t) > 0$.*

An intuitive motivation for the requirement in Remark 4.2 is that we need to move through a path of plausible samples $\hat{\boldsymbol{x}}_t$ at every step $t$. One simple way to guarantee this is by adding a small amount of noise to $\boldsymbol{y}$. Then $p(\boldsymbol{y})$, and therefore $p(\boldsymbol{x}_t)$ will be non-zero everywhere. More discussion about this is presented at the end of this section, but first, we present a toy example to motivate our approach.

**A Toy Example:** Let us assume we observe noisy samples $\boldsymbol{y} = \boldsymbol{H}\boldsymbol{x} + \boldsymbol{n}$, where $\boldsymbol{n} \sim \mathcal{N}(0, \sigma^2 Id)$, drawn from a discrete multimodal distribution $p(\boldsymbol{x}) = \sum_{i=1}^d w_i \delta_{\boldsymbol{x}-\boldsymbol{c}_i}$, where $\boldsymbol{c}_i \in \mathbb{R}^N$ and $w_i \geq 0$ and $\sum_{i=1}^d w_i = 1$.

Let $\boldsymbol{x}_t$ be the intermediate degraded samples according to equation 1. In this simple example, there is a closed form expression for all posterior distributions and conditional means; namely,

$$p(\boldsymbol{x}_t|\boldsymbol{x}) = G\left(\frac{\boldsymbol{x}_t - \boldsymbol{H}_t\boldsymbol{x}}{\sigma_t}\right) \quad \text{and} \quad p(\boldsymbol{x}_t) = \sum_{i=1}^d w_i G\left(\frac{\boldsymbol{x}_t - \boldsymbol{H}_t\boldsymbol{c}_i}{\sigma_t}\right), \tag{3}$$

where $\boldsymbol{H}_t = (1-t)\boldsymbol{I} + t\boldsymbol{H}$ and $G(\boldsymbol{x})$ is a Gaussian kernel with identity covariance and $\sigma_t = t\sigma$. Then,

$$\mathbb{E}_{\boldsymbol{x}\sim p(\boldsymbol{x}|\boldsymbol{x}_t)}[\boldsymbol{x}] = \int \frac{p(\boldsymbol{x}_t|\boldsymbol{x})p(\boldsymbol{x})}{p(\boldsymbol{x}_t)}\boldsymbol{x}d\boldsymbol{x} = \frac{\sum_{i=1}^d \boldsymbol{c}_i w_i G\left(\frac{\boldsymbol{x}_t-\boldsymbol{H}_t\boldsymbol{c}_i}{\sigma_t}\right)}{\sum_{i=1}^d w_i G\left(\frac{\boldsymbol{x}_t-\boldsymbol{H}_t\boldsymbol{c}_i}{\sigma_t}\right)}$$

The iteration from equation 2 becomes:

$$\hat{\boldsymbol{x}}_{t-\delta} = \frac{\delta}{t} \frac{\sum_{i=1}^{d} \boldsymbol{c}_i w_i G\left(\frac{\hat{\boldsymbol{x}}_t - \boldsymbol{H}_t \boldsymbol{c}_i}{\sigma_t}\right)}{\sum_{i=1}^{d} w_i G\left(\frac{\hat{\boldsymbol{x}}_t - \boldsymbol{H}_t \boldsymbol{c}_i}{\sigma_t}\right)} + \left(1 - \frac{\delta}{t}\right) \hat{\boldsymbol{x}}_t.$$

Figure 1 shows the results of applying the iterative regression scheme given by the above equation in two different examples. The iterative regression converges to one of the four possible modes (shown in orange), while the regression to the mean is always a weighted average of all possible modes (i.e., a blurry reconstruction, shown in red).

**Training and Inference:** InDI ideal iterative scheme given by equation 2 requires to compute an estimate of the clean image at every step $t$ (i.e., $\mathbb{E}[\boldsymbol{x}_0 \mid \boldsymbol{x}_t]$). To that extent, we train a family of regressors $F_\theta(\cdot; t)$, each of them specialized in reconstructing $\boldsymbol{x}_0$ from $\boldsymbol{x}_t$ at a given $t$. That is,

$$\min_\theta \mathbb{E}_{\boldsymbol{x}, \boldsymbol{y} \sim p(\boldsymbol{x}, \boldsymbol{y})} \mathbb{E}_{t \sim p(t)} \| F_\theta(\boldsymbol{x}_t, t) - \boldsymbol{x} \|_p, \tag{4}$$

where $p(t)$ is a predefined distribution for $t$ (e.g., uniform). The model $F_\theta$ allows us to do incremental reconstruction where from time step $t$ we predict the slightly less corrupted signal at time $t - \delta$ as given in equation 2. Thus, the iterative scheme becomes:

$$\hat{\boldsymbol{x}}_{t-\delta} = \frac{\delta}{t} F_\theta(\hat{\boldsymbol{x}}_t, t) + \left(1 - \frac{\delta}{t}\right) \hat{\boldsymbol{x}}_t, \tag{5}$$

where $0 < \delta \leq 1$. Although $\delta$ could be a function of time, in practice we use a constant time step, $\delta = \frac{1}{N}$, where $N$ is the number of steps.

**InDI as a Residual Flow ODE:** In the limit, as $\delta \to 0$, equation 5 leads to an ordinary differential equation (ODE); namely

$$\frac{d\boldsymbol{x}_t}{dt} = \lim_{\delta \to 0} \frac{\boldsymbol{x}_t - \boldsymbol{x}_{t-\delta}}{\delta} = \frac{\boldsymbol{x}_t - F_\theta(\boldsymbol{x}_t, t)}{t}, \tag{6}$$

where in the ideal case $F_\theta(\boldsymbol{x}_t, t) = \mathbb{E}[\boldsymbol{x}_0 \mid \boldsymbol{x}_t]$. The ODE can be interpreted as a "residual flow" because the right-hand side is the (normalized) residual of the inversion process at time $t$. We are interested in the solution of this equation at $t = 0$, starting from the initial condition $\boldsymbol{x}_1 = \boldsymbol{y}$ at $t = 1$. The residual flow formulation can be used to develop other numerical procedures using standard ODE solvers. Exploring this is left to future work.

Another use of the continuous formulation is to understand the behavior of the proposed iterative procedure in terms of concrete examples. In Appendix B we show how the residual flow can be used to analyze the specific case where the prior is Gaussian and the restoration task is denoising.

**Connection to Denoising Score-Matching and Probabilistic ODE:** An interesting connection emerges when the degradation is Gaussian noise (standard deviation $\sigma^2$). In this case, InDI's ODE in equation 6 boils down to the score-matching probabilistic ODE of Song et al. (2021b). More specifically, let $\boldsymbol{x}_t = \boldsymbol{x} + t\boldsymbol{n}$, so the noise level in $\boldsymbol{x}_t$ is $\sigma_t^2 = t^2 \sigma^2$. The probabilistic flow ODE (Eq(13) in Song et al. (2021b)) is given by

$$\frac{d\boldsymbol{x}_t}{dt} = -\frac{1}{2} \frac{d(\sigma_t^2)}{dt} \nabla_{\boldsymbol{x}_t} \log p_t(\boldsymbol{x}_t) = -t\sigma^2 \nabla_{\boldsymbol{x}_t} \log p_t(\boldsymbol{x}_t).$$

According to the denoising score-matching (DSM) approximation (Vincent (2011)),

$$-\nabla_{\boldsymbol{x}_t} \log p_t(\boldsymbol{x}_t) \approx \frac{\boldsymbol{x}_t - F_\theta(\boldsymbol{x}_t, t)}{\sigma_t^2} = \frac{\boldsymbol{x}_t - F_\theta(\boldsymbol{x}_t, t)}{t^2 \sigma^2},$$

so we end up recovering precisely the same ODE as in InDI. Furthermore, when $F_\theta(\boldsymbol{x}_t, t) = \mathbb{E}[\boldsymbol{x}_0 \mid \boldsymbol{x}_t]$ (i.e., MMSE estimator) the DSM approximation is exact and the relation is given by Tweedie's formula (Robbins, 1956; Efron, 2011).

---

**Algorithm 1** Iterative Image Restoration (Inference)

---

**Require:** $\boldsymbol{y}, F_\theta(\cdot, t), \delta, \epsilon_t$
    $\boldsymbol{n} \sim \mathcal{N}(\boldsymbol{0}, \boldsymbol{I})$
    $\hat{\boldsymbol{x}}_1 = \boldsymbol{y} + \epsilon_1 \boldsymbol{n}$
    **for** $t = 1$ **to** $0$ **with step** $-\delta$ **do**
        $\boldsymbol{\zeta} \sim \mathcal{N}(\boldsymbol{0}, \boldsymbol{I})$
        $\hat{\boldsymbol{x}}_{t-\delta} \leftarrow \frac{\delta}{t} F_\theta(\hat{\boldsymbol{x}}_t, t) + \left(1 - \frac{\delta}{t}\right) \hat{\boldsymbol{x}}_t + (t - \delta)\sqrt{\epsilon_{t-\delta}^2 - \epsilon_t^2}\boldsymbol{\zeta}$         ▷ Update rule from equation 10.
    **end for**
    **return** $\hat{\boldsymbol{x}}_0$

---

**Stochastic Perturbation:** To make sure we have the regularity requirements for the iterative procedure from equation 2 to be well defined (Remark 4.2), we add a small amount of white noise to the low-quality input. As shown in Section 5 this leads to a significant improvement in image quality in certain tasks (in particular those that are restorations from deterministic degradations).

Our model with this noise perturbation becomes:

$$\boldsymbol{x}_t = (1 - t)\boldsymbol{x} + t\boldsymbol{y}' = (1 - t)\boldsymbol{x} + t\boldsymbol{y} + t\epsilon\boldsymbol{n} \quad \text{with } t \in [0, 1], \tag{7}$$

where $\boldsymbol{y}' = \boldsymbol{y} + \epsilon\boldsymbol{n}$, $\epsilon$ is a small constant (e.g., $\epsilon = 0.01$, where image values are in $[-1, 1]$), and $\boldsymbol{n} \sim \mathcal{N}(0, Id)$.

A slightly more general formulation incorporates the perturbation as a general Brownian motion, where we can explicitly control the level of noise at each step. That is,

$$\boldsymbol{x}_t = (1 - t)\boldsymbol{x} + t\boldsymbol{y} + \sqrt{t}\epsilon_t\boldsymbol{\eta}_t \quad \text{with } t \in [0, 1], \tag{8}$$

where $\epsilon_t$ is a non-negative function, and $\boldsymbol{\eta}_t$ is the standard Brownian motion having zero mean and covariance $t\boldsymbol{I}$ at index $t$.

In this more general setting, the base training objective becomes

$$\min_\theta \mathbb{E}_{\boldsymbol{x}, \boldsymbol{y} \sim p(\boldsymbol{x}, \boldsymbol{y})} \mathbb{E}_{t \sim p(t)} \mathbb{E}_{\boldsymbol{n} \sim \mathcal{N}(0, Id)} \left\| F_\theta\left((1 - t)\boldsymbol{x} + t\boldsymbol{y} + t\epsilon_t\boldsymbol{n}; t\right) - \boldsymbol{x} \right\|_p. \tag{9}$$

And as a result, the general inference procedure in equation 2 becomes:

$$\hat{\boldsymbol{x}}_{t-\delta} = \frac{\delta}{t} F_\theta(\hat{\boldsymbol{x}}_t, t) + \left(1 - \frac{\delta}{t}\right) \hat{\boldsymbol{x}}_t + (t - \delta)\sqrt{\epsilon_{t-\delta}^2 - \epsilon_t^2}\boldsymbol{\zeta}, \tag{10}$$

where the reconstruction process starts from $t = 1$, $\hat{\boldsymbol{x}}_1 = \boldsymbol{y} + \epsilon\boldsymbol{n}$ and $\boldsymbol{n} \sim \mathcal{N}(0, Id)$. At each step, a new $\boldsymbol{\zeta} \sim \mathcal{N}(0, Id)$ is sampled and noise is added to the current state. The added Gaussian noise is such that the noise at time $t$ has variance $t^2 \epsilon_t^2$ as required by equation 8. To be well defined $\epsilon_t$ needs to be a non-negative non-increasing function of $t$. In the limit case where $\epsilon_t = \epsilon$ we are in the simplified case given by equation 7, while if $\epsilon_t = \frac{\epsilon}{\sqrt{t}}$ the noise perturbation is a pure Brownian motion.

Our full iterative restoration inference scheme is given in Algorithm 1.

## 5 Experiments

We train and evaluate our framework on four widely popular image restoration tasks: motion deblurring, defocus deblurring, compression artifacts removal and single image super-resolution. Our formulation is generative-based, and we show that can be used for image generation even if this is not the main focus of the present work. To evaluate the quality of the proposed method we compute several distortion based and perceptual metrics: PSNR, LPIPS (Zhang et al., 2018), FID (Fréchet Inception Distance) (Heusel et al., 2017), and KID (Kernel Inception Distance) (Bińkowski et al., 2018).

**Perception–Distortion tradeoff (Blau & Michaeli, 2018):** To illustrate the potential of the method, we present results when using different number of steps for the reconstruction. This has a direct impact on

the perception–distortion tradeoff. In general, a single step reconstruction with our model, will lead to an estimate that minimizes the average point distortion (e.g., PSNR) but this can be only done to the detriment of the perceptual quality.

**Model Architecture and Training:** We adopt a U-Net-like architecture similar to the ones in diffusion strategies (Saharia et al., 2021; Whang et al., 2022). Following Whang et al. (2022) we removed attention layers and group normalization to have a fully-convolutional architecture. The size of the model varies for each evaluated task (in general we chose a model size proportional to the size of the dataset to avoid significant overfitting). The model is trained on image crops using ADAM optimizer. Learning rates and other hyper-parameters are given in Appendix C. For each experiment, we train a single model $F_\theta(\cdot; t)$ that is conditioned on the parameter $t$. The model is trained using the loss function of equation 9 with $p = 1$. We found that the distribution of $t$, $p(t)$ plays an important role. In Section 6.3 we present an empirical analysis of its impact.

## 5.1 Motion Deblurring

Motion deblurring is a very challenging restoration task. Motion is intrinsically random in the sense that a priori we don't have a known degradation model. The current best end-to-end deep learning solution is to train regression models using paired data sharp, blurry frames. One of the most adopted training datasets is the GoPro motion deblurring dataset (Nah et al., 2017) containing 3214 pairs of clean and blurry $1280 \times 720$ images (1111 are reserved for evaluation). The blurry frames are generated by recording high-frame rate video clips and then averaging consecutive frames to simulate blurs caused due to longer exposure. We follow the standard setup (Nah et al., 2017; Kupyn et al., 2019; Chen et al., 2021a; Cho et al., 2021; Suin et al., 2020; Zhang et al., 2019) and perform training data augmentation with random horizontal/vertical flips and 90/180/270 rotations. We did not introduce additional noise to the blurry inputs ($\epsilon = 0$ in equation 7).

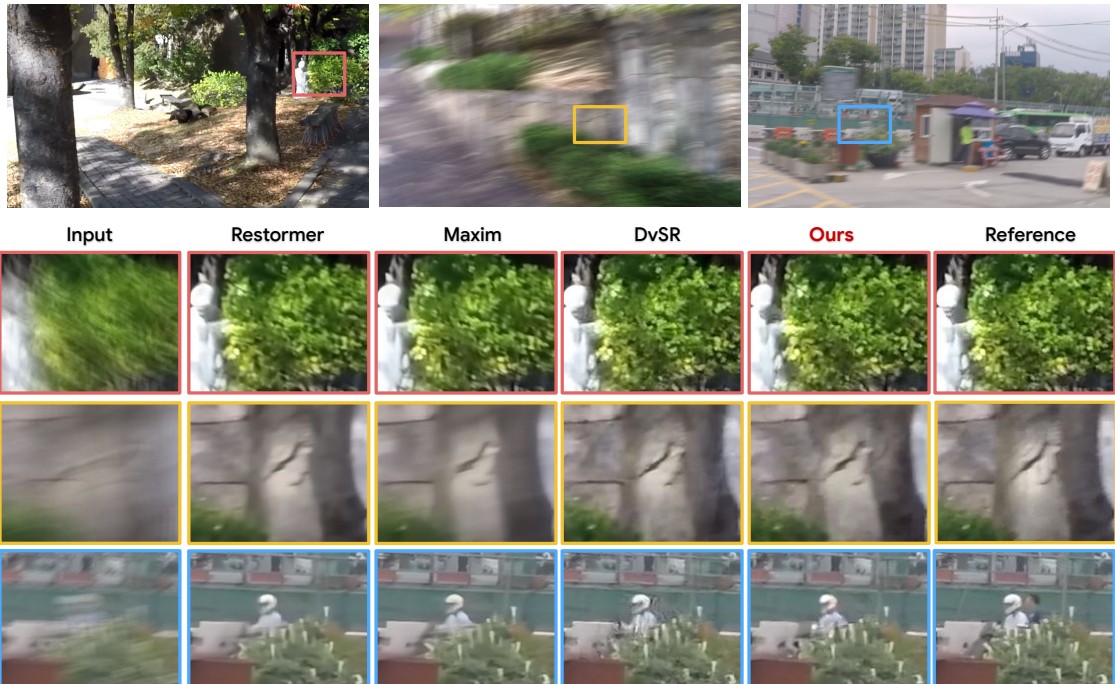

Figure 2: Examples of deblurred images from GoPro dataset. Our iterative reconstruction leads achieves better reconstruction of detailed textures than regression based models (Restormer, Maxim) and similar quality than conditional DPMs (DvSR). More results are provided in Appendix.

Figure 2 shows a visual comparison of our iterative image restoration and current state-of-the-art deblurring models. The iterative scheme produces images with much more details than regression based solutions (Restormer (Zamir et al., 2022), MAXIM (Tu et al., 2022)). Our results are similar to the ones generated by

Table 1: Image motion deblurring on the GoPro (Nah et al., 2017) dataset. Best values and second-best values for each metric are color-coded. KID values are scaled by a factor of 1000 for readability.

| | Perceptual | | | | Distortion | |
|---|---|---|---|---|---|---|
| | LPIPS↓ | NIQE↓ | FID↓ | KID↓ | PSNR↑ | SSIM↑ |
| Ground Truth | 0.0 | 3.21 | 0.0 | 0.0 | ∞ | 1.000 |
| HINet (Chen et al., 2021a) | 0.088 | 4.01 | 17.91 | 8.15 | 32.77 | 0.960 |
| MPRNet (Zamir et al., 2021) | 0.089 | 4.09 | 20.18 | 9.10 | 32.66 | 0.959 |
| MIMO-UNet+ (Cho et al., 2021) | 0.091 | 4.03 | 18.05 | 8.17 | 32.45 | 0.957 |
| SAPHNet (Suin et al., 2020) | 0.101 | 3.99 | 19.06 | 8.48 | 31.89 | 0.953 |
| DeblurGANv2 (Kupyn et al., 2019) | 0.117 | 3.68 | 13.40 | 4.41 | 29.08 | 0.918 |
| DvSR (Whang et al., 2022) | 0.059 | 3.39 | 4.04 | 0.98 | 31.66 | 0.948 |
| DvSR-SA (Whang et al., 2022) | 0.078 | 4.07 | 17.46 | 8.03 | 33.23 | 0.963 |
| Restormer (Zamir et al., 2022) | 0.084 | 4.11 | 19.33 | 8.78 | 32.92 | 0.961 |
| MAXIM (Tu et al., 2022) | 0.087 | 3.94 | 22.76 | 10.06 | 32.86 | 0.962 |
| NAFNet (Chen et al., 2022) | 0.078 | 4.07 | 17.87 | 8.27 | 33.71 | 0.967 |
| **Ours (10 steps)** | 0.058 | 3.32 | 3.55 | 0.56 | 31.49 | 0.946 |

current conditional diffusion models (DvSR (Whang et al., 2022)). Quantitative results on the GoPro dataset are presented in Table 1. The proposed iterative reconstruction procedure achieves a new state-of-the-art performance across perceptual metrics while maintaining competitive PSNR to existing methods.

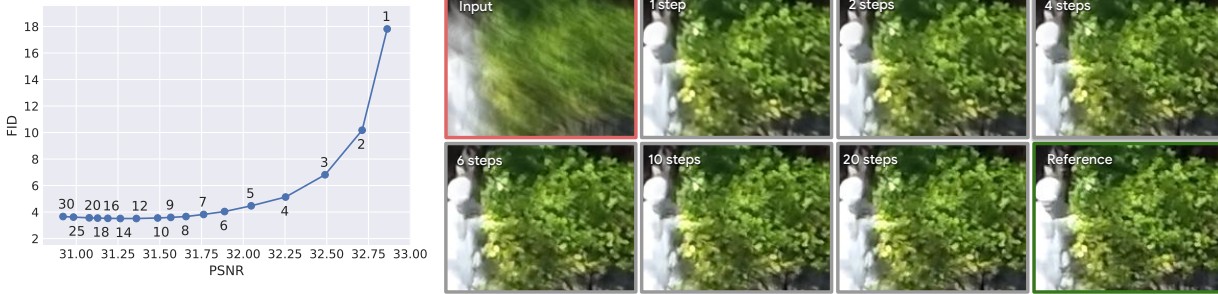

Figure 3: Number of steps. The total number of steps in the iterative regression has a direct impact on the quality. The number of steps seems to control the Perception-distortion tradeoff. One step leads to the best possible MSE reconstruction (minimum distortion) but large perceptual discrepancy.

**Number of steps.** Figure 3 shows the impact of the number of inference steps on the Perception–Distortion trade-off (Blau & Michaeli, 2018). While doing a reconstruction on a single step (e.g., direct regression) produces the best PSNR, the perceptual metrics are significantly improved when the number of steps is larger than one. Both metrics can't be optimized simultaneously (Blau & Michaeli, 2018).

## 5.2 Single-Image Super-resolution

We evaluated the iterative restoration methodology on single-image $4\times$ super-resolution on the `div2k` dataset (Agustsson & Timofte, 2017). This dataset contains 1000 2K-resolution images (800 for training, 100 images for validation, 100 testing). We compare to other state-of-the art models that span from regression models having powerful architectures (Wang et al., 2018; Chen et al., 2021b; Liang et al., 2022) and/or generative formulations: GAN based, i.e., LDL (Liang et al., 2022), ESRGAN (Wang et al., 2018), BSRGAN (Zhang et al., 2021); and also based on Normalizing Flows, SRFLOW (Lugmayr et al., 2020).

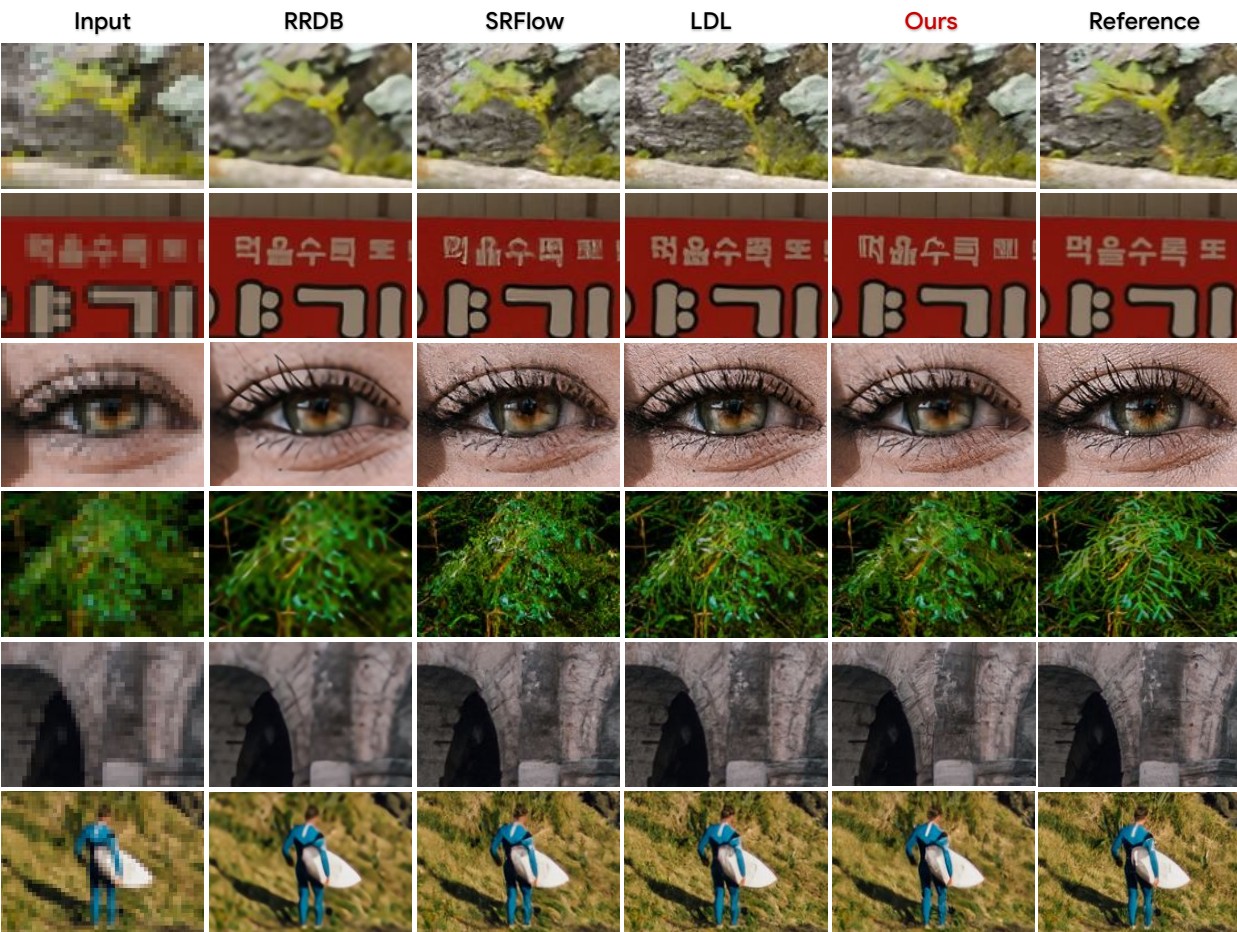

Figure 4: Examples image 4× upscaling. Our iterative reconstruction leads achieves better reconstruction of detailed textures than RRDB regression model (Wang et al., 2018), less high-frequency artifacts than SRFlow (Lugmayr et al., 2020) generative normalizing flow, and comparable visual quality than LDL (Liang et al., 2022) a state-of-the-art customized generative adversarial model. More results are given in Appendix.

Figure 5(b) summarizes the quantitative results on 4× SR div2k validation dataset. Figure 4 shows a selection of results. Our proposed framework leads to upscaled images with more defined structure than regression based formulations producing larger PSNR, e.g., RRDB (Wang et al., 2018). The recently introduced adversarial formulation LDL (Liang et al., 2022) produces slightly better fine grain details. This could indicate that in the situation where there is limited training data, careful adversarial formulation may be more data efficient.

**The importance of adding noise in deterministic super-resolution.** In our formulation of super-resolution, the degradation is a deterministic linear (blurring plus subsampling) operator. Figure 5(a) shows the importance of adding a small amount of noise to the input image. Directly applying the original iterative procedure (without adding noise to the input) leads to a blurry reconstruction (high PSNR but low FID score, $\epsilon = 0.0$ in Figure 5 (a)). Adding a small amount of noise ($\epsilon > 0$ in Figure 5(a)) leads to significant better results in terms of perceptual quality (e.g., FID score).

### 5.3 Defocus deblurring

Defocus deblurring is the task of reducing the blur due to limited depth-of-field or misfocus. For such purposes we used the Canon dual-pixel (DP) defocus dataset (DDPD) provided by Abuolaim & Brown

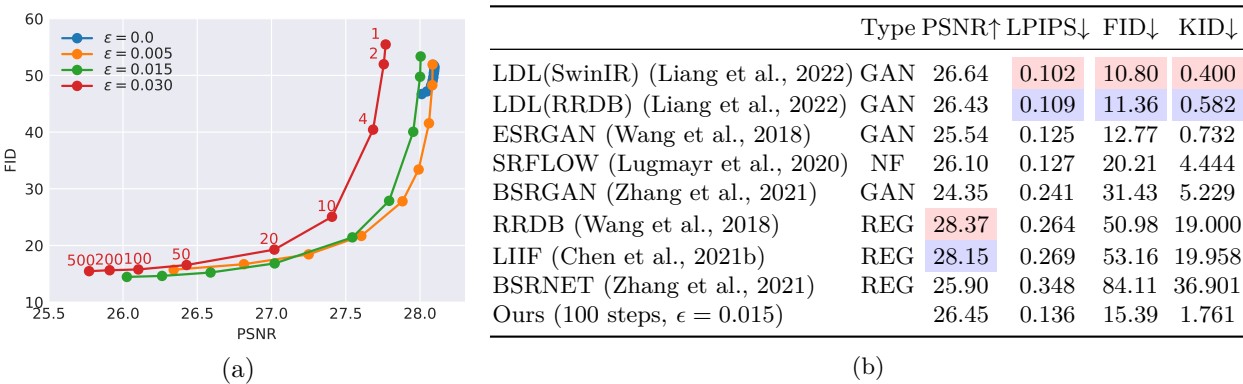

| | Type | PSNR↑ | LPIPS↓ | FID↓ | KID↓ |
|---|---|---|---|---|---|
| LDL(SwinIR) (Liang et al., 2022) | GAN | 26.64 | 0.102 | 10.80 | 0.400 |
| LDL(RRDB) (Liang et al., 2022) | GAN | 26.43 | 0.109 | 11.36 | 0.582 |
| ESRGAN (Wang et al., 2018) | GAN | 25.54 | 0.125 | 12.77 | 0.732 |
| SRFLOW (Lugmayr et al., 2020) | NF | 26.10 | 0.127 | 20.21 | 4.444 |
| BSRGAN (Zhang et al., 2021) | GAN | 24.35 | 0.241 | 31.43 | 5.229 |
| RRDB (Wang et al., 2018) | REG | 28.37 | 0.264 | 50.98 | 19.000 |
| LIIF (Chen et al., 2021b) | REG | 28.15 | 0.269 | 53.16 | 19.958 |
| BSRNET (Zhang et al., 2021) | REG | 25.90 | 0.348 | 84.11 | 36.901 |
| Ours (100 steps, $\epsilon = 0.015$) | | 26.45 | 0.136 | 15.39 | 1.761 |

(a)                  (b)

Figure 5: 4× Super-resolution on div2k dataset (Agustsson & Timofte, 2017). Best values and second-best values for each metric are color-coded

(2020), and train a defocus deblurring model only using single image input (i.e., we don't use the dual-pixel images given in the dataset). The DDPD dataset contains 1000 pairs of sharp and blurry $6720 \times 4480$ images, of which 30% are reserved for validation and testing. The blurry/sharp frames are generated by capturing two consecutive snapshots by changing the camera parameters (lens aperture). high-frame rate video clips and then averaging consecutive frames to simulate blurs caused due to longer exposure.

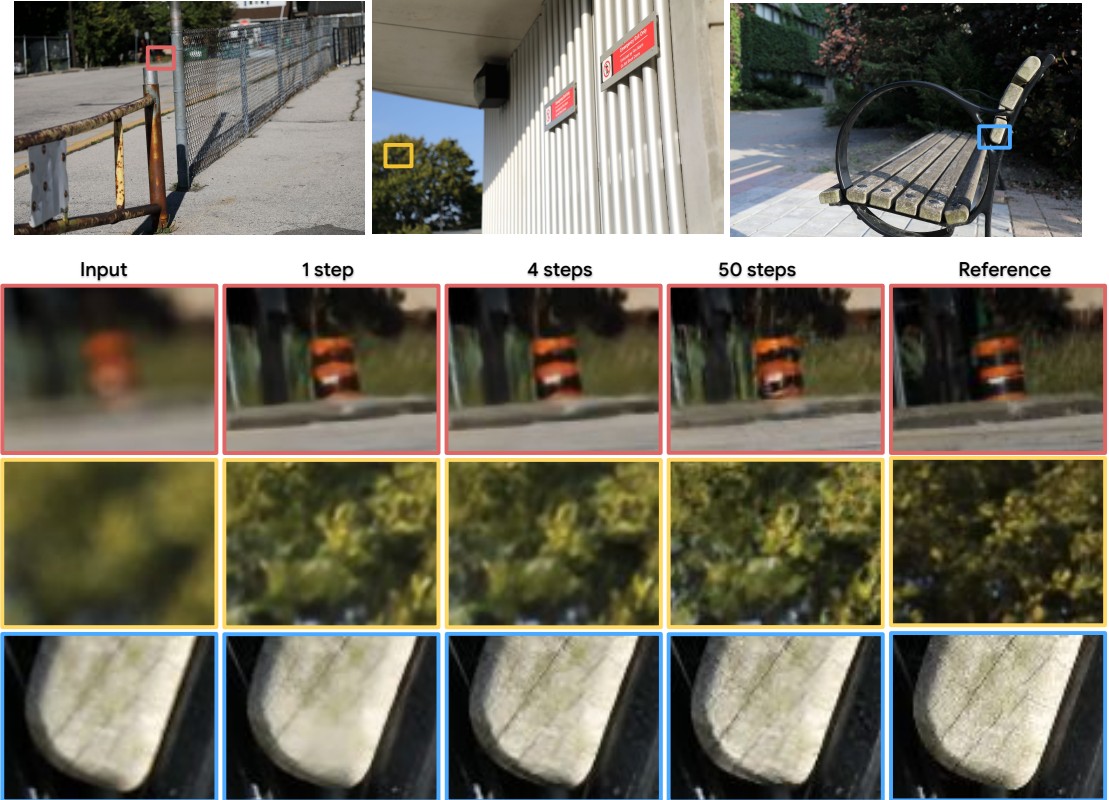

Figure 6: Examples of restored images from defocus DDPD dataset. The iterative reconstruction leads to images with more texture compared to the direct regression (1 step). More results are provided in Appendix.

Figure 6 shows a visual comparison of our iterative image restoration when a different number of inference steps is used. Increasing the number of steps has a direct impact on the quality of the result. Quantitative results on the DDPD dataset are summarized in Table 3 in Appendix. As in the other experiments the best

PSNR is obtained with a single step (direct regression), while the best perceptual metrics are obtained when the restoration is done in multiple steps. We did not introduce additional noise to the blurry inputs ($\epsilon = 0$ in equation 7).

### 5.4 Compression artifact removal

JPEG compression introduces blocking artifacts and lack of high-frequency details. We evaluated the proposed method on the task of removing strong JPEG compression artifacts (quality factor 15). To generate the training data we use the 1000 `div2k` high-quality images (Agustsson & Timofte, 2017). We evaluated the model on div2k validation set.

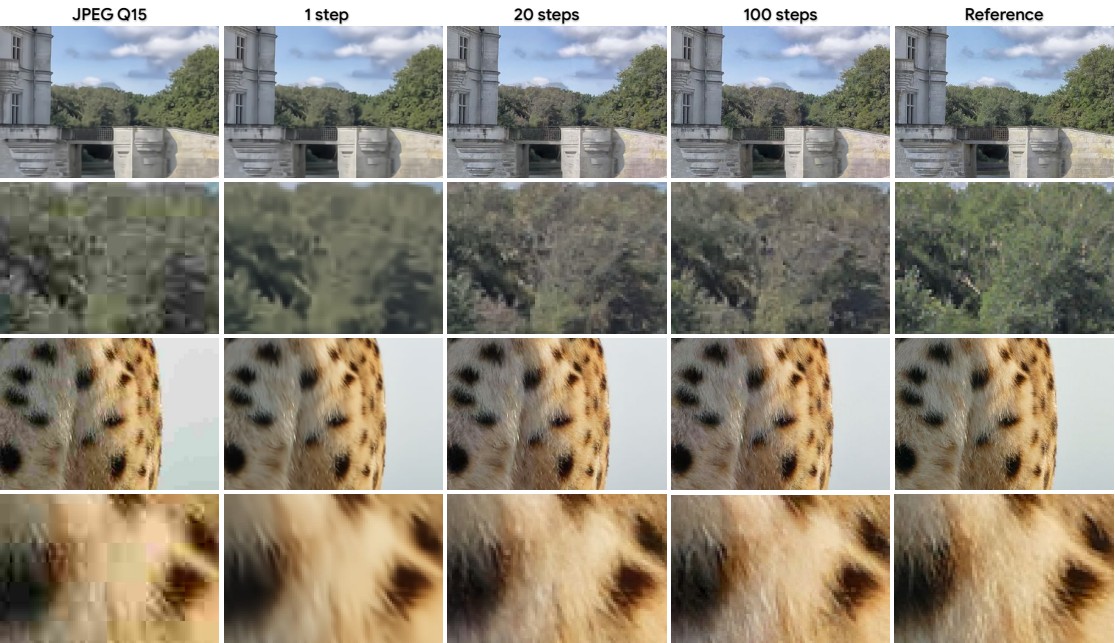

Figure 7: JPEG Compression artifacts removal (quality factor 15). As more inference steps are used more details are generated.

Figure 7 show some visual results of restored images with the model applying a different number of steps. As more inference steps are used the restored images have more details. More results on JPEG compression removal are discussed in the next section.

## 6 Discussion

### 6.1 A generative framework

A natural question to ask is whether the proposed approach is also generative in the spirit of diffusion formulations. Namely, if we take our formulation to the limit where the low-quality image is fully degraded, then could we potentially generate new samples from scratch?

To test the idea we trained a restoration model that starts from pure Gaussian noise paired to a $64 \times 64$ celebA image and then proceed as described above. Figure 8 shows some generated samples with this formulation. The generated samples have a FID=9.19, which is not state-of-the-art[2] but illustrate the point. In this specific case, our proposed methodology leads to a similar denoising training loss as the one in DDPM (Ho et al., 2020). Despite this similarity, the two methods come from different motivations/formulations and therefore have different inference strategies. We didn't fine-tune architecture or hyper-parameters to boost

---

[2]It is competitive with other methods from a couple years ago

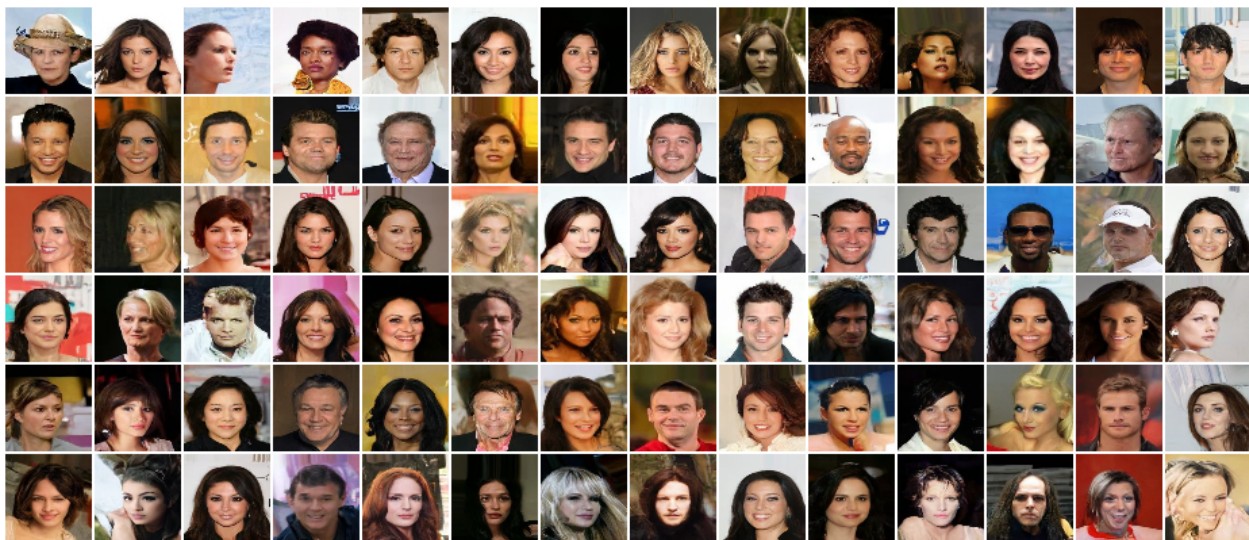

Figure 8: Examples of CelebA $64 \times 64$ generated samples (FID=9.19, 150 steps). Our iterative reconstruction can be potentially used to generated samples from a fully-degraded image (noise).

the performance since our goal was to present the idea and show that the formulation, at its core, can be generative as well.

## 6.2 Comparison of Inference Algorithms

In what follows we discuss different alternatives for recovering the clean sample with the trained models.

**Naive Procedure and Relevance to Cold Diffusion:** Given equation 1, one may be tempted to directly replace the clean image by the current estimate. This would lead to $\hat{\boldsymbol{x}}_t = (1-t)F_\theta(\hat{\boldsymbol{x}}_s, t) + t\boldsymbol{y}$, and the inference iterative rule would become

$$\hat{\boldsymbol{x}}_{t-\delta} = (1 - t + \delta)F_\theta(\hat{\boldsymbol{x}}_t, t) + (t - \delta)\boldsymbol{y}. \tag{11}$$

Cold Diffusion (Bansal et al., 2022) proposes to generate images by inverting an arbitrary *known* degradation $D(\boldsymbol{x}, s)$, where $s$ controls the strength. Our formulation is more general in the sense that we don't require an explicit knowledge of $D$. To apply Cold Diffusion sampling in our context, we define $D(\boldsymbol{x}, s) = (1-s)\boldsymbol{x} + s\boldsymbol{y}$ (given by equation 1). This leads to Cold Diffusion's naive sampling (Algorithm 1 in Bansal et al. (2022)),

$$\hat{\boldsymbol{x}}_{t-\delta} = D(F_\theta(\hat{\boldsymbol{x}}_t, t), t - \delta) = (1 - t + \delta)F_\theta(\hat{\boldsymbol{x}}_t, t) + (t - \delta)\boldsymbol{y}. \tag{12}$$

Note that this sampling scheme is the same as the one in Eq. 11. Cold diffusion improved sampling (Algorithm 2 in Bansal et al. (2022)) is given by,

$$\hat{\boldsymbol{x}}_{t-\delta} = \hat{\boldsymbol{x}}_t - D(F_\theta(\hat{\boldsymbol{x}}_t, t), t) + D(F_\theta(\hat{\boldsymbol{x}}_t, t), t - \delta) \tag{13}$$

$$= \hat{\boldsymbol{x}}_t - (1 - t)F_\theta(\hat{\boldsymbol{x}}_t, t) - t\boldsymbol{y} + (1 - t + \delta)F_\theta(\hat{\boldsymbol{x}}_t, t) + (t - \delta)\boldsymbol{y} \tag{14}$$

$$= \hat{\boldsymbol{x}}_t + \delta(F_\theta(\hat{\boldsymbol{x}}_t, t) - \boldsymbol{y}). \tag{15}$$

In Figure 9 we compare our inference algorithm (equation 5), the naive inference algorithm (equation 11), and our adaptation of the Cold Diffusion sampler to our formulation (equation 15). In general, the naive sampler produces good results with very few steps (N=2,3) but then diverges. Our adaptation of Cold Diffusion sampler produces competitive results, while leading to slightly worse FID scores for the same distortion level than our proposed algorithm. In the limit, as the number of steps becomes very large, Cold Diffusion sampler seems to converge to a stable point, while ours after a certain large number of steps, deteriorates.

Figure 9. shows the FID score of CelebA 64x64 generated images when using the three different variants of the inference algorithm (sampler).

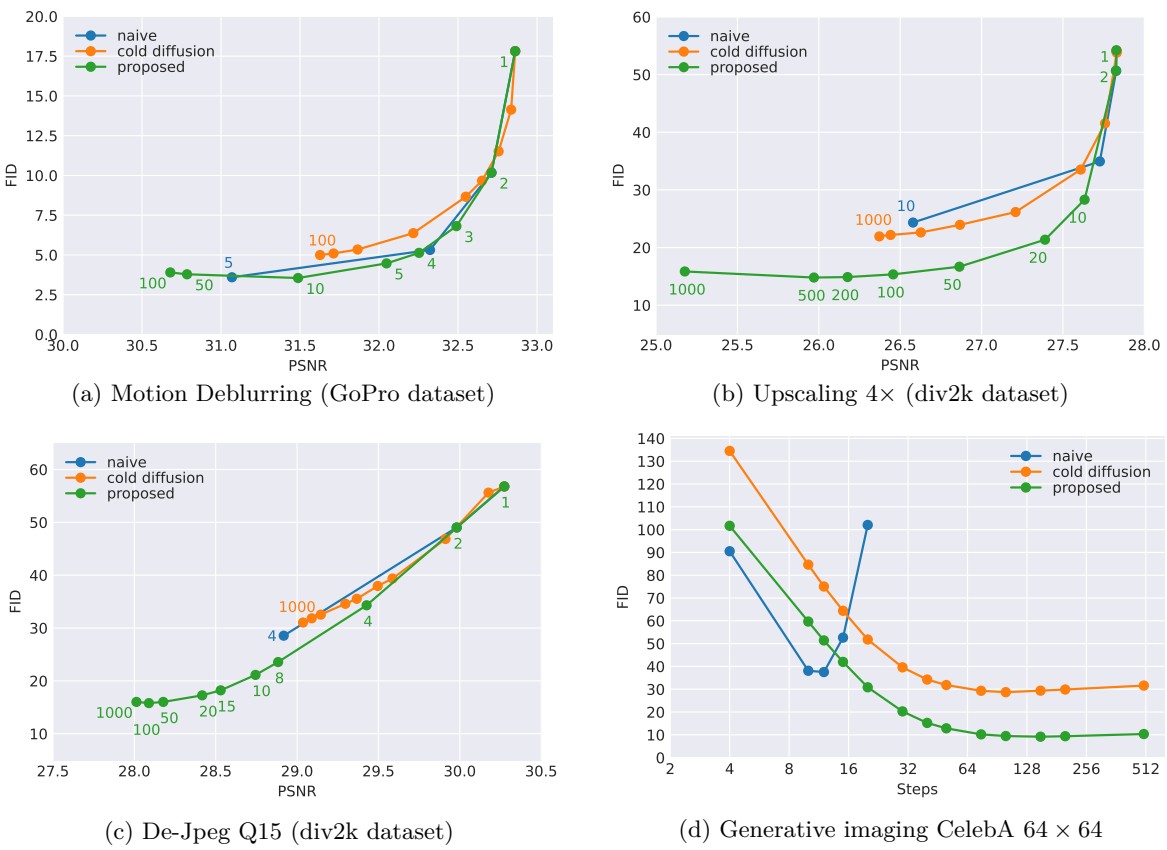

(a) Motion Deblurring (GoPro dataset)

(b) Upscaling 4× (div2k dataset)

(c) De-Jpeg Q15 (div2k dataset)

(d) Generative imaging CelebA $64 \times 64$

Figure 9: Impact of the Inference Algorithm for some of the different tested tasks. The naive inference algorithm (equation 11) produces a good baseline when used with very few steps (i.e. 2–5) but then diverges. The proposed updated rule (equation 5) produces better results than the one adapted from Cold Diffusion (Bansal et al., 2022) given in (equation 15). At very large number of steps, Cold Diffusion seems to be the most stable of the three tested samplers. Each of the points in each shown curve represents the results with a different number of steps (similar to what is shown in Figure 3. Points that leads to values that are out-of-the shown region are left out for improving visualization.

## 6.3 Impact of distribution p(t)

The impact of the distribution of $t$ used during training has a clear impact on performance. We evaluated several different options that are summarized in Figure 10 (a). Figure 10 (b) shows the results when different distributions are adopted. The best results are obtained when the model is trained with a bias towards $t = 1$ (more degradation). Intuitively, this could imply that the iterative procedure needs to be more certain of the direction to move at the very early steps of the procedure. Nonetheless, the best distribution can depend on a combination of model capacity and restoration task so we are not drawing general conclusions.

## 6.4 Impact of adding noise on inverting deterministic degradations

JPEG compression is a non-linear, but deterministic, degradation. We empirically verified that adding a small amount of noise helps to improve the results as shown in Figure 11. We tested the variant of the inference algorithm that adds noise at each step (so the noise becomes a Brownian motion, e.g., $\epsilon_t = \epsilon/\sqrt{t}$), and adding a constant noise level at the initial step ($\epsilon_t = \epsilon$). We did not observe any practical difference in the two approaches.

| key | $p(t)$ | description |
|---|---|---|
| `'linear_0'` | $t \sim \mathcal{U}[0,1]$ | Uniform distribution |
| `'linear_a'` | $t \sim (1-a)\mathcal{U}[0,1] + a\delta_1$, where $a < 1$ | Uniform distribution w/bias to $t=1$ |
| `'bias_t1'` | $t = g(s)$, $s \sim \mathcal{U}[0,1]$ and $g(s) = \sin(s\pi/2)$ | Sine based distribution (bias to $t=1$) |
| `'bias_t0'` | $t = g(s)$, $s \sim \mathcal{U}[0,1]$ and $g(s) = \sin((s-1)\pi/2) + 1$ | Sine based distribution (bias to $t=0$) |
| `'bias_t0_t1'` | $t = g(s)$, with $s \sim \mathcal{U}[0,1]$ and $g(s) = \sin(s\pi/2)^2$ | Sine based distribution (bias to $t=1$) |

(a) Different evaluated $p(t)$ training distributions.

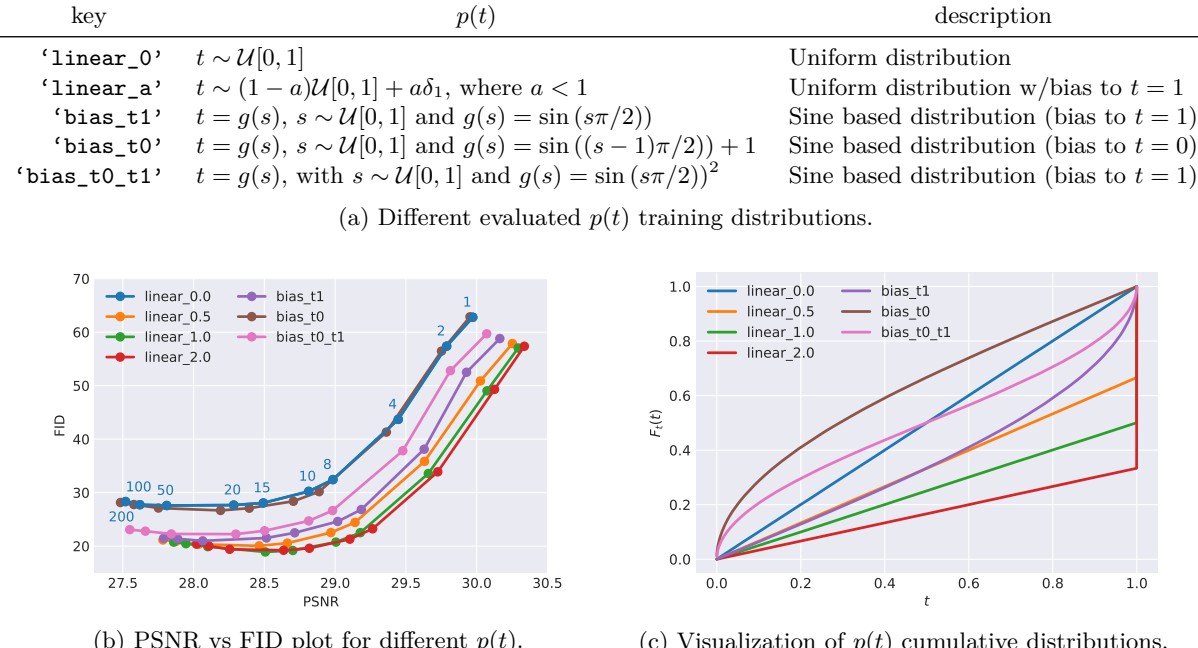

(b) PSNR vs FID plot for different $p(t)$.

(c) Visualization of $p(t)$ cumulative distributions.

Figure 10: Impact of the distribution of $p(t)$ during training. Results for JPEG Compression removal (Q=15). The best results are obtained when the distribution of $t$ is biased towards $t=1$.

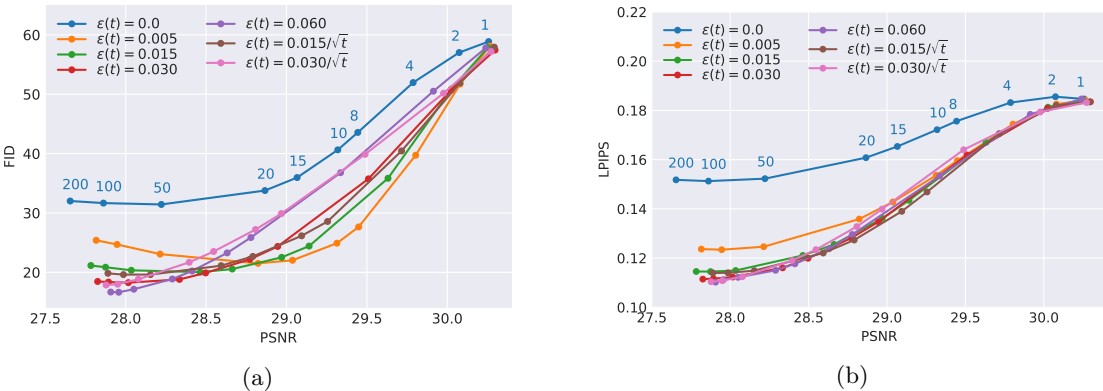

(a)

(b)

Figure 11: Impact of adding noise to the low-quality input in JPEG compression removal.

## 6.5 Comparison to a Conditional Denoising Diffusion Model

We compare InDI to a vanilla conditional DDPM (Ho et al., 2020). We trained a vanilla conditional DDPM, using the (continuous) noise level as an additional input, similarly as done in Saharia et al. (2021); Whang et al. (2022). The model architecture is the same as in InDI but the auxiliary noise image (needed in any DDPM) is concatenated with the low-quality input at each step. Figure 12 shows a comparison between InDI and the conditional DDPM. To generate the DDPM plot we merged several possible noise schedules using different number of steps that span the perception-distortion tradeoff. InDI produces comparable results using significant less number of steps than the vanilla DDPM.

## 7 Conclusions and Limitations

We presented a novel formulation of image restoration that circumvents the regression-to-the mean problem. This allows us to get restored images with superior realism and perceptual quality, while still having a low

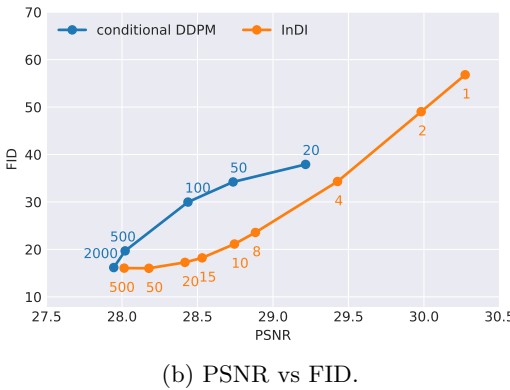

(b) PSNR vs FID.

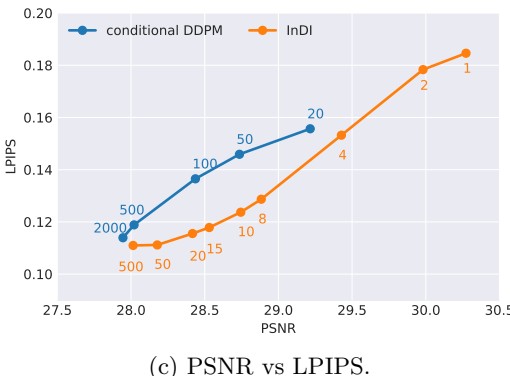

(c) PSNR vs LPIPS.

Figure 12: Comparison of InDI to a conditional Denoising Diffusion Probabilistic Model (DDPM). The DDPM requires a noise schedule for inference. The plot showed herein was built by fusioning six different noise schedules taking the best PSNR vs FID score at a given number of steps. Our proposed InDI algorithm produces comparable results in much less number of steps.

distortion error. Our method is motivated by the observation that restoration from a small distortion is a better-conditioned problem. We therefore break a restoration task into many small ones – each of them easier (and less ill-posed) than the larger problem we solve overall. This enables our iterative approach to transforming the degraded image into a high-quality image, in spirit similar to current generative diffusion models.

**Limitations.** The present formulation is a supervised one, requiring paired training data. As such, for each type of degradation we need to train a specialized model, in contrast to unsupervised formulations such as RED (Romano et al., 2017), PnP (Venkatakrishnan et al., 2013; Kamilov et al., 2022), or DDRM (Kawar et al., 2022). Additionally, given the dependence to paired training data, its performance for out-of-distribution samples is not guaranteed. This question requires more in-depth analysis. Finally, while the proposed iterative inference algorithm produces high-quality restorations, in some tasks performance degrades after a certain number of steps. This is likely due to the accumulation of errors, and will likely require a more robust inference scheme.

For future work, we would like to better characterize the limiting points of the proposed inference procedure. Other possible research avenues are developing robust formulations that can successfully handle out-of-domain input.

### Acknowledgments

The authors would like to thank our colleagues Jon Barron, Tim Salimans, Jascha Sohl-dickstein, Ben Poole, José Lezama, Sergey Ioffe, and Jason Baldridge for helpful discussions.

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

# A  Proof of Proposition 4.1

*Proof.* We have $\boldsymbol{x}_s = (1-s)\boldsymbol{x} + s\boldsymbol{y}$, and $\boldsymbol{x}_t = (1-t)\boldsymbol{x} + t\boldsymbol{y}$, so by substituting $\boldsymbol{y}$ from one to the other we get,

$$\boldsymbol{x}_s = (1-s)\boldsymbol{x} + s\left(\frac{\boldsymbol{x}_t - (1-t)\boldsymbol{x}}{t}\right) \tag{16}$$

$$= \boldsymbol{x} - s\boldsymbol{x} + \frac{s}{t}\boldsymbol{x}_t - \frac{s}{t}\boldsymbol{x} + s\boldsymbol{x} \tag{17}$$

$$= \left(1 - \frac{s}{t}\right)\boldsymbol{x} + \frac{s}{t}\boldsymbol{x}_t. \tag{18}$$

Then,

$$\mathbb{E}[\,\boldsymbol{x}_s\,|\,\boldsymbol{x}_t\,] = \int \boldsymbol{x}_s p_{\boldsymbol{x}_s|\boldsymbol{x}_t}(\boldsymbol{x}_s|\boldsymbol{x}_t)d\boldsymbol{x}_s \tag{19}$$

$$= \int \boldsymbol{x}_s p_{\boldsymbol{x}|\boldsymbol{x}_t}\left(\frac{t\boldsymbol{x}_s - s\boldsymbol{x}_t}{t-s}\Big|\boldsymbol{x}_t\right)\frac{t}{t-s}d\boldsymbol{x}_s \tag{20}$$

$$= \int \frac{(t-s)\boldsymbol{x} + s\boldsymbol{x}_t}{t}p_{\boldsymbol{x}|\boldsymbol{x}_t}(\boldsymbol{x}|\boldsymbol{x}_t)d\boldsymbol{x} \tag{21}$$

$$= \left(1 - \frac{s}{t}\right)\int \boldsymbol{x}p_{\boldsymbol{x}|\boldsymbol{x}_t}(\boldsymbol{x}|\boldsymbol{x}_t)d\boldsymbol{x} + \frac{s}{t}\boldsymbol{x}_t \tag{22}$$

$$= \left(1 - \frac{s}{t}\right)\mathbb{E}[\,\boldsymbol{x}\,|\,\boldsymbol{x}_t\,] + \frac{s}{t}\boldsymbol{x}_t. \tag{23}$$

where we have applied the fact that $p_{\boldsymbol{x}_s|\boldsymbol{x}_t}(\boldsymbol{x}_s|\boldsymbol{x}_t) = p_{\boldsymbol{x}|\boldsymbol{x}_t}(\boldsymbol{x}|\boldsymbol{x}_t)\frac{t}{t-s}$ and $\boldsymbol{x} = \frac{t\boldsymbol{x}_s - s\boldsymbol{x}_t}{t-s}$. $\qquad\square$

# B  Denoising with a Gaussian Prior

Let's analyze InDI's behavior in the particular case where $p(\mathbf{x}) = N(\mathbf{c}, \sigma_c^2\mathbf{I})$, and the restoration task is denoising $\mathbf{y} = \mathbf{x} + \mathbf{n}$, where $\mathbf{n}$ is white Gaussian of fixed standard deviation $\sigma_N$. Then, $p(\mathbf{y}|\mathbf{x}) = N(\mathbf{x}, \sigma_N^2\mathbf{I})$ and $\mathbb{E}[\mathbf{x}|\mathbf{x}_t] = \frac{\sigma_c^2\mathbf{x}_t + t^2\sigma_N^2\mathbf{c}}{\sigma_c^2 + t^2\sigma_N^2}$, where we have used the fact that $\mathbf{x}_t = (1-t)\mathbf{x} + t\mathbf{y} = \mathbf{x} + t\mathbf{n}$.

InDI's ideal ODE is given by $\frac{d\mathbf{x}_t}{dt} = \frac{\mathbf{x}_t - \mathbb{E}[\mathbf{x}|\mathbf{x}_t]}{t}$, which in this case becomes:

$$\frac{d\mathbf{x}_t}{dt} = \frac{t\sigma_N^2(\mathbf{x}_t - \mathbf{c})}{\sigma_N^2 t^2 + \sigma_c^2}.$$

We are interested in solving this equation at $t = 0$, with boundary condition $\mathbf{x}_1 = \mathbf{y}$ at $t = 1$. This is a separable ODE having general solution: $\mathbf{x}_t = \mathbf{c} + (\mathbf{y} - \mathbf{c})\sqrt{\frac{t^2 + \alpha^2}{1 + \alpha^2}}$, where $\alpha = \frac{\sigma_c}{\sigma_N}$. The solution at $t = 0$, is

$$\mathbf{x}_{\text{InDI}} = \mathbf{c} + (\mathbf{y} - \mathbf{c})\sqrt{\frac{\sigma_c^2}{\sigma_c^2 + \sigma_N^2}}.$$

Note that $\mathbb{E}[\mathbf{x}_{\text{InDI}}] = \mathbf{c} = \mathbb{E}[\mathbf{x}]$, and the covariance $\text{cov}(\mathbf{x}_{\text{InDI}}) = \text{cov}(\mathbf{y})\frac{\sigma_c^2}{\sigma_c^2 + \sigma_N^2} = \text{cov}(\mathbf{x})$, since $\text{cov}(\mathbf{y}) = (\sigma_c^2 + \sigma_N^2)\mathbf{I}$. Given $p(\mathbf{x})$ and $p(\mathbf{x}_{\text{InDI}})$ are Gaussian distributions with same mean and covariance, we have $p(\mathbf{x}_{\text{InDI}}) = p(\mathbf{x})$. That is, in the limit, InDI generates samples from the prior distribution $p(\mathbf{x})$.

It is worth noting that in this case the MMSE and MAP estimates coincide,

$$\mathbf{x}_{\text{MMSE}} = \mathbf{x}_{\text{MAP}} = \frac{\sigma_c^2\mathbf{y} + \sigma_N^2\mathbf{c}}{\sigma_c^2 + \sigma_N^2},$$

and are in fact different from InDI's estimate.

## C  Model and Training Details

In all our restoration experiments we use a U-Net-like architecture (Ronneberger et al., 2015) similar to the one in SR3 (Saharia et al., 2021) and DvSR (Whang et al., 2022). We followed the same adaptations as the ones introduced in Whang et al. (2022) to make it fully-convolutional (removed self-attention layers and group normalization). Our U-Net has an adaptive number of resolutions each of them having an arbitrary number of channels (given by a multiplication factor from a base set of channels).

Table 2 summarizes the model definition for each of the tested applications.

Table 2: Model and training parameters for each restoration task.

| | channels | multipliers | noise | $p(t)$ | batch size | learning rate | crop size | # params. |
|---|---|---|---|---|---|---|---|---|
| motion deblurring | 64 | [1,2,3,4] | $\epsilon = 0$ | `bias_t1` | 256 | $10^{-4}$ | $128 \times 128$ | 27.68M |
| defocus deblurring | 64 | [1,2,4,4] | $\epsilon = 0$ | `linear_0.5` | 1024 | $10^{-4}$ | $128 \times 128$ | 33.57M |
| JPEG restoration | 64 | [1,2,4,4] | $\epsilon = 0.060$ | `linear_1.0` | 1024 | $10^{-4}$ | $128 \times 128$ | 33.57M |
| 4× super-resolution | 96 | [1,2,3,4] | $\epsilon = 0.015$ | `bias_t1` | 256 | $10^{-4}$ | $256 \times 256$ | 62.25M |

All models are trained for 500K steps using 32 TPUv3 cores. We used the Adam optimizer with a fixed learning rate, and EMA decay rate of 0.9999. Models were trained using the respective indicated distribution for $p(t)$. For the super-resolution model, low-resolution crops of size $64 \times 64$ are upscaled using bilinear interpolation to $256 \times 256$ before feeding them into the model.

## D  Additional Results

Table 3: Defocus Deblurring on the DDPD dataset (Abuolaim & Brown, 2020). Best values and second-best values for each metric are color-coded. KID values are scaled by a factor of 1000 for readability

| Steps | PSNR | LPIPS | FID | KID |
|---|---|---|---|---|
| 1 | 24.75 | 0.206 | 40.80 | 16.72 |
| 2 | 24.74 | 0.201 | 23.29 | 6.18 |
| 4 | 24.55 | 0.195 | 17.92 | 3.52 |
| 10 | 24.24 | 0.188 | 15.68 | 2.19 |
| 50 | 23.89 | 0.186 | 16.12 | 2.48 |
| 100 | 23.82 | 0.187 | 16.34 | 2.63 |
| 500 | 23.77 | 0.189 | 16.53 | 2.49 |

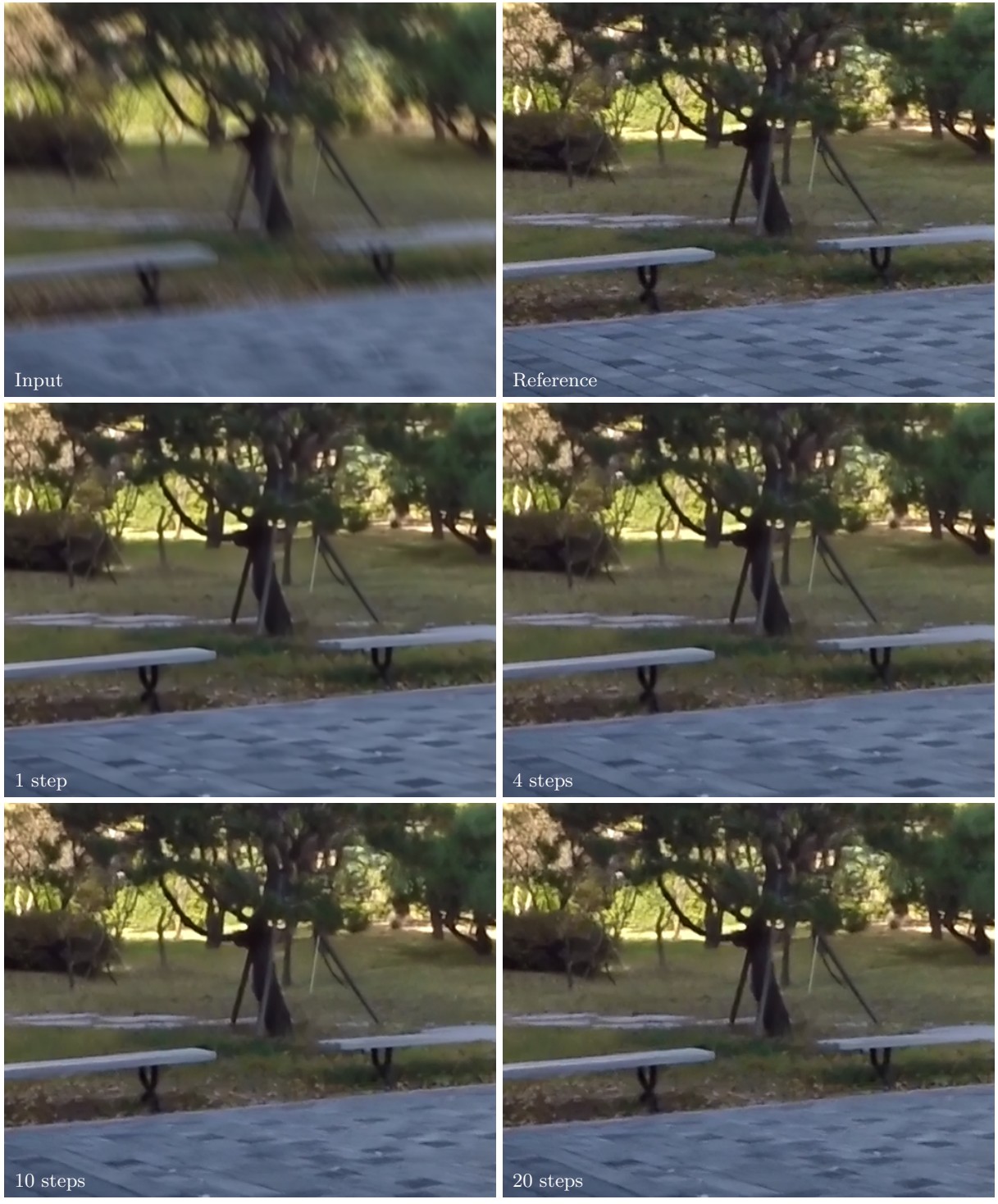

Figure 13: Additional GoPro deblurring results. The proposed method (InDI) applied with different number of reconstruction steps. Best viewed electronically.

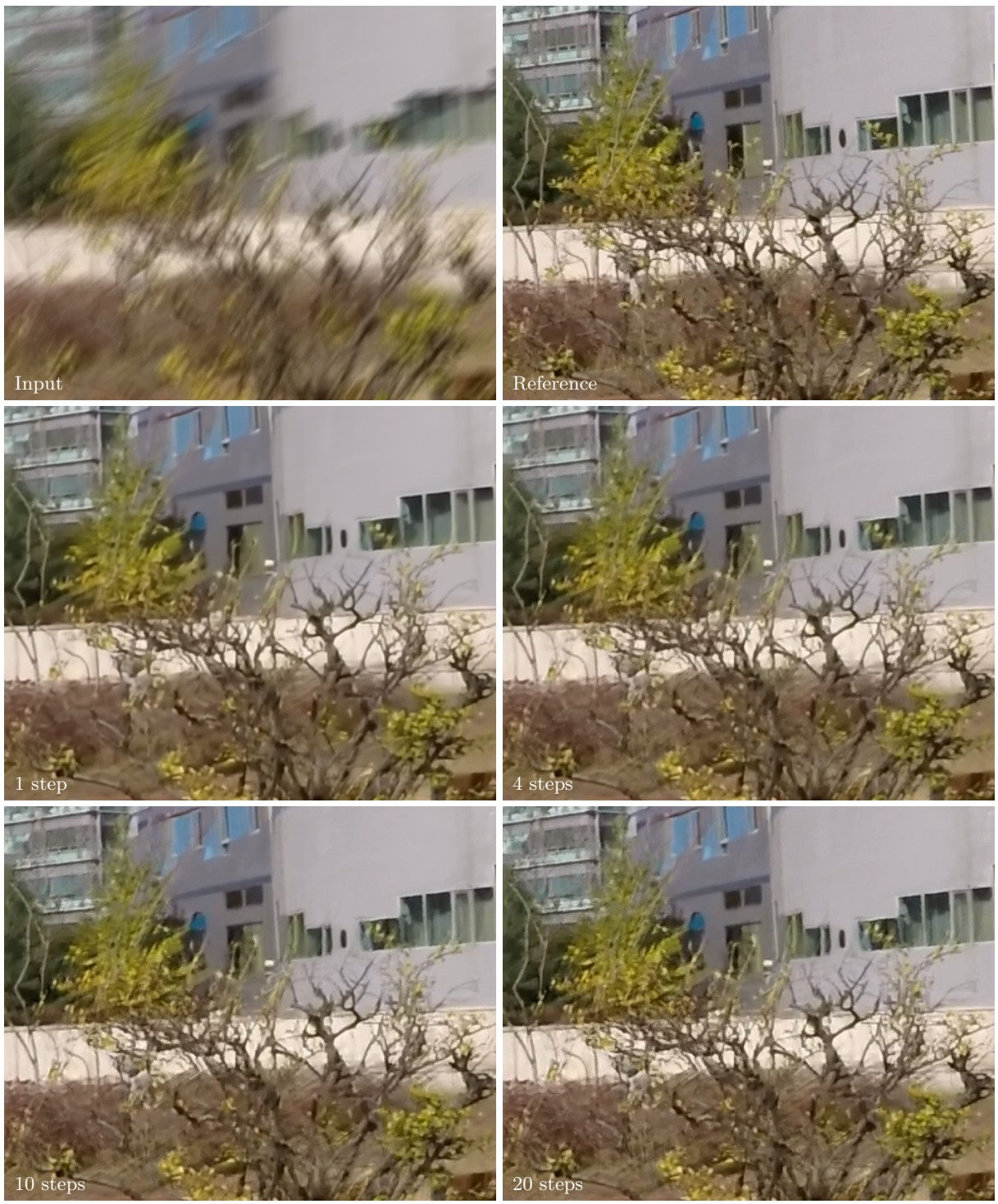

Figure 14: Additional GoPro deblurring results. The proposed method (InDI) applied with different number of reconstruction steps. Best viewed electronically.

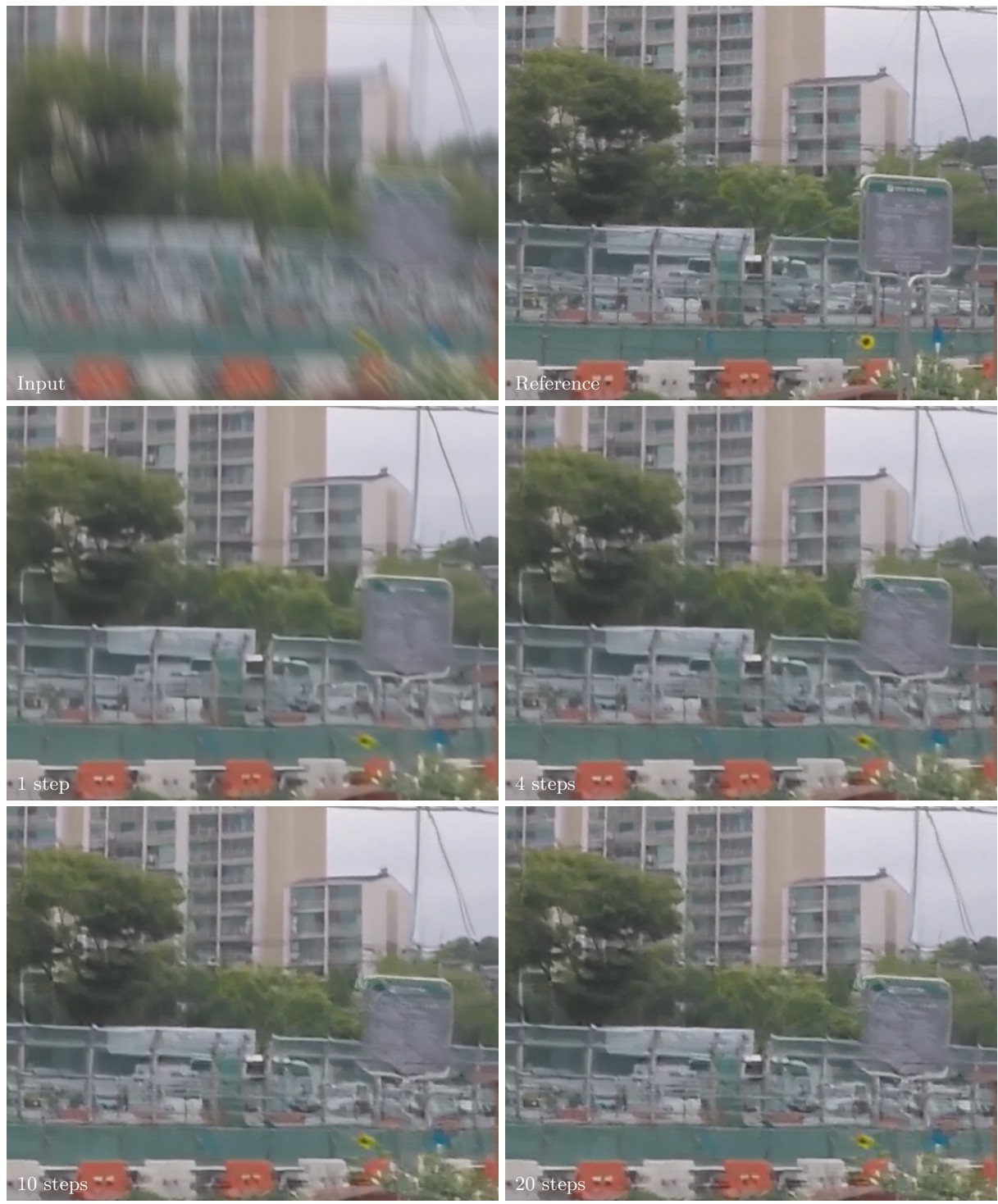

Figure 15: Additional GoPro deblurring results. The proposed method (InDI) applied with different number of reconstruction steps. Best viewed electronically.

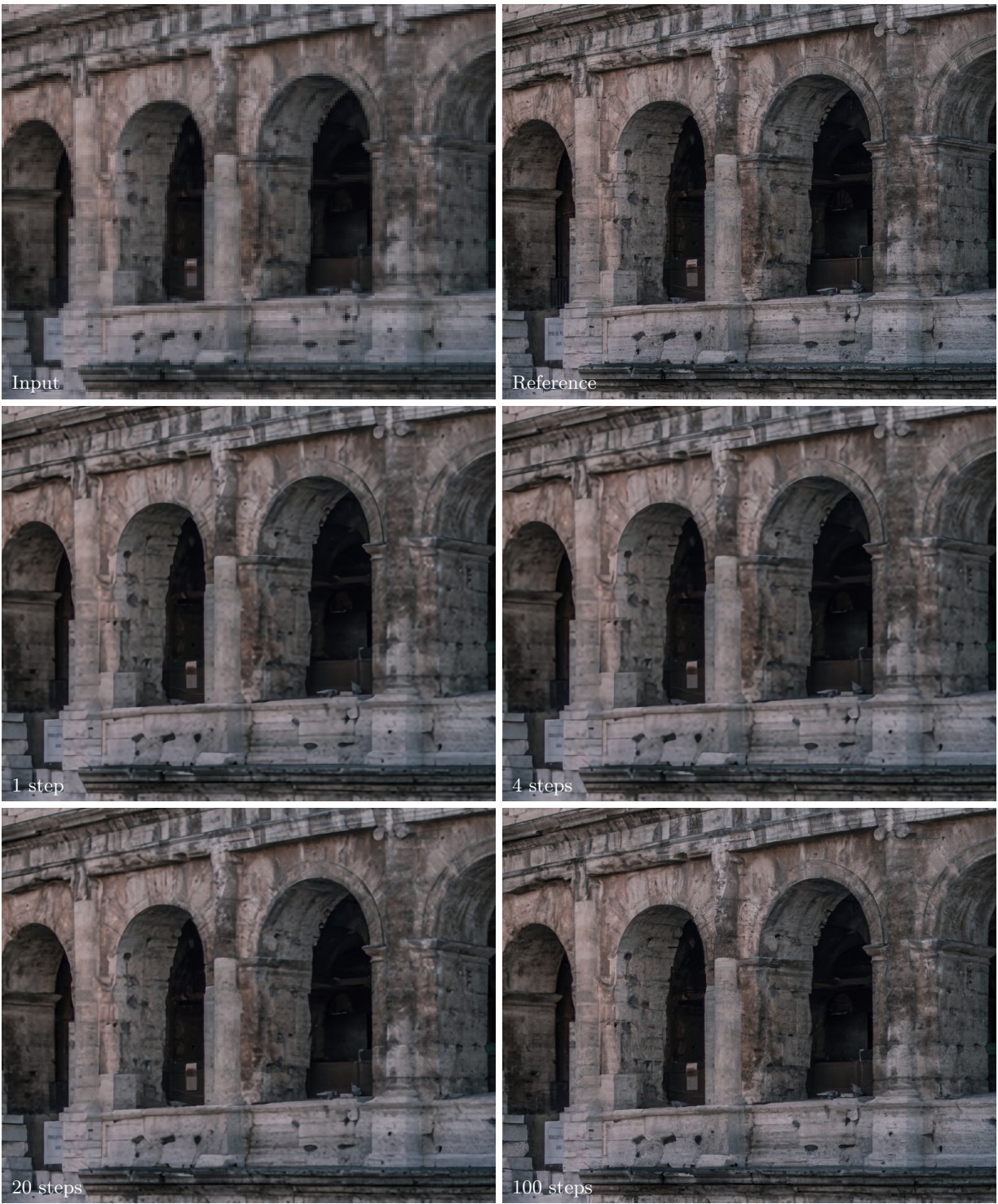

Figure 16: 4× super-resolution results (div2k dataset). The proposed method (InDI) applied with different number of reconstruction steps. Best viewed electronically.

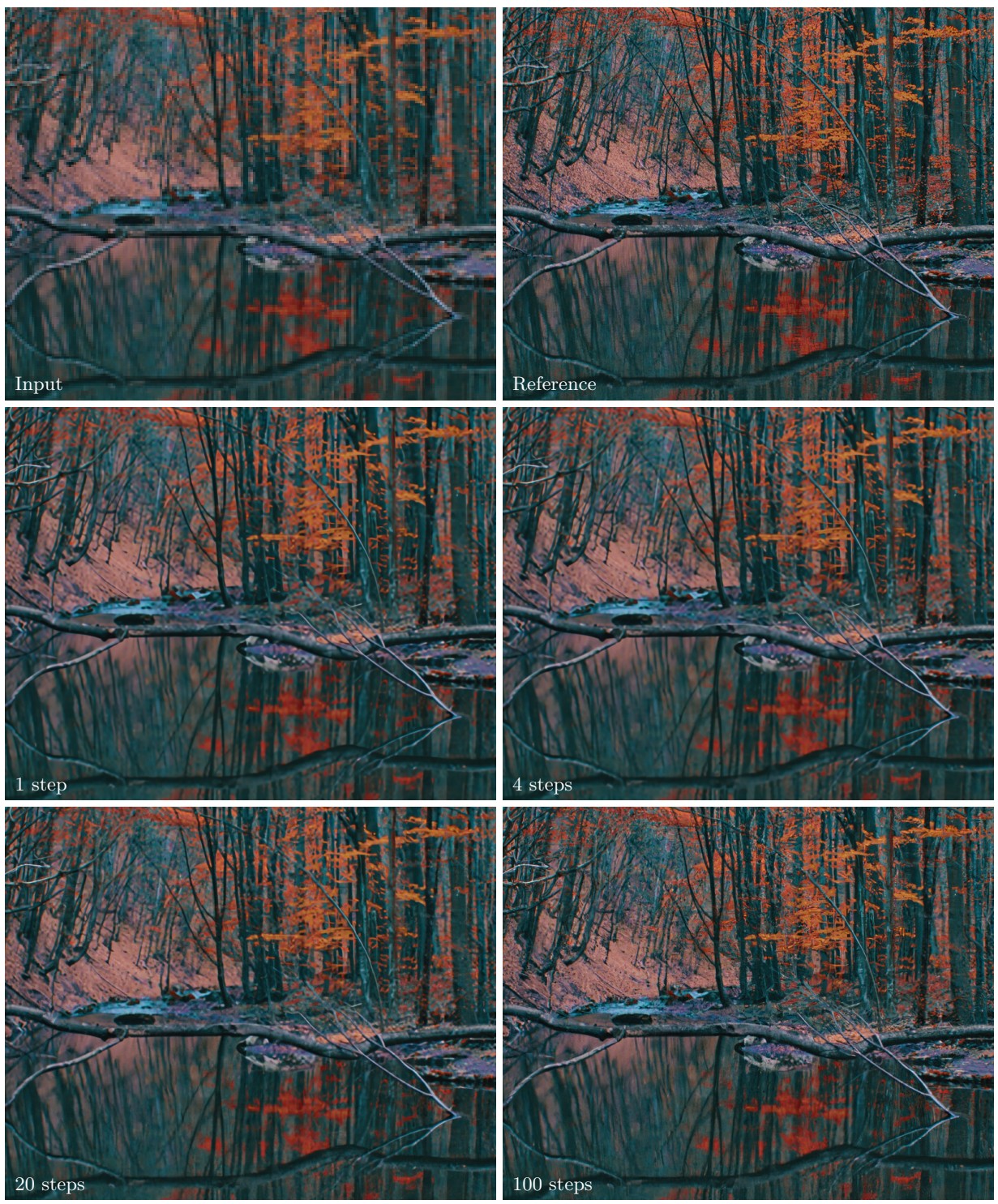

Figure 17: 4× super-resolution results (div2k dataset). The proposed method (InDI) applied with different number of reconstruction steps. Best viewed electronically.

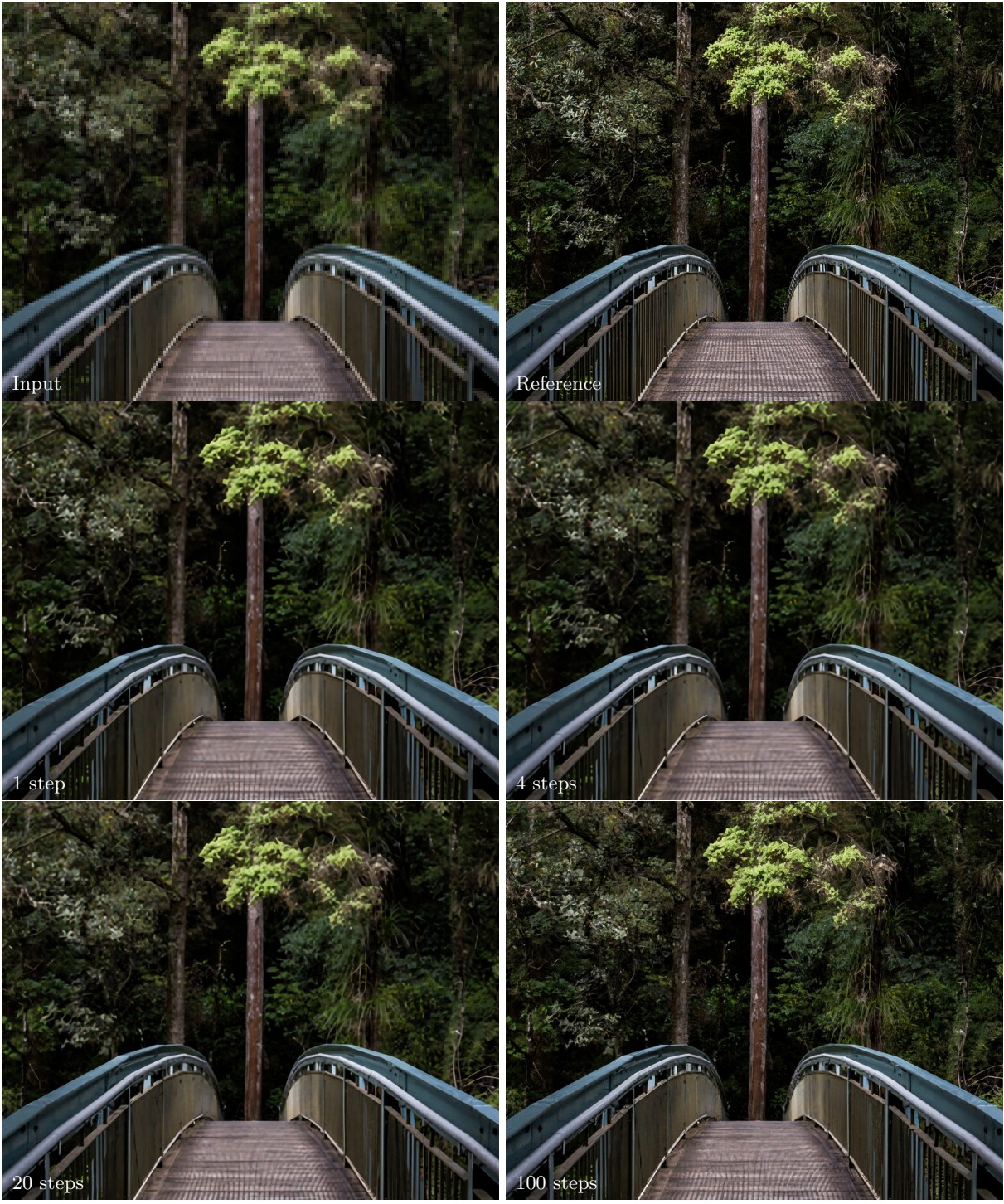

Figure 18: 4× super-resolution results (div2k dataset). The proposed method (InDI) applied with different number of reconstruction steps. Best viewed electronically.

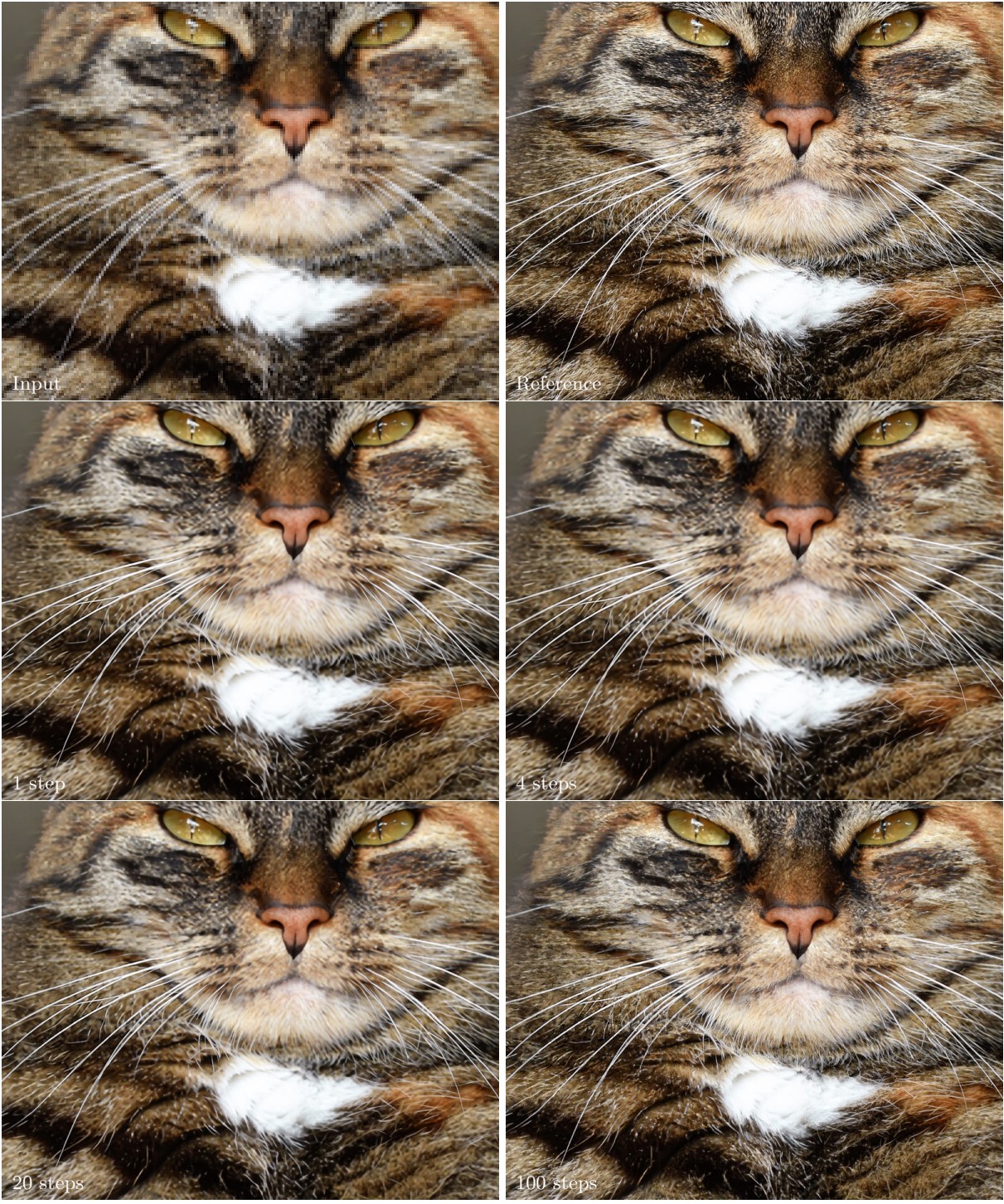

Figure 19: 4× super-resolution results (div2k dataset). The proposed method (InDI) applied with different number of reconstruction steps. Best viewed electronically.

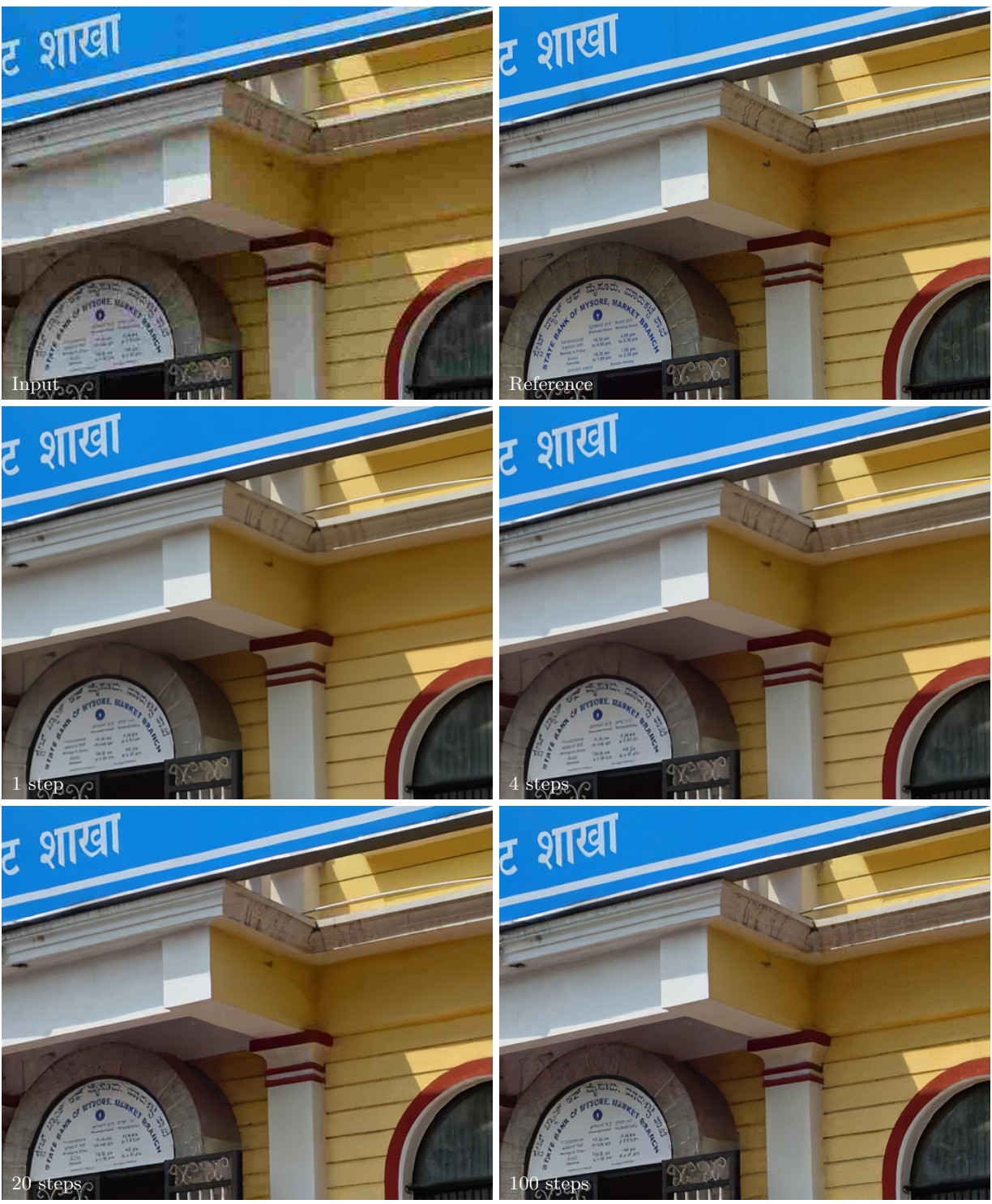

Figure 20: JPEG compression artifact removal results (quality factor 15, div2k test dataset). The proposed method (InDI) applied with different number of reconstruction steps. Best viewed electronically.

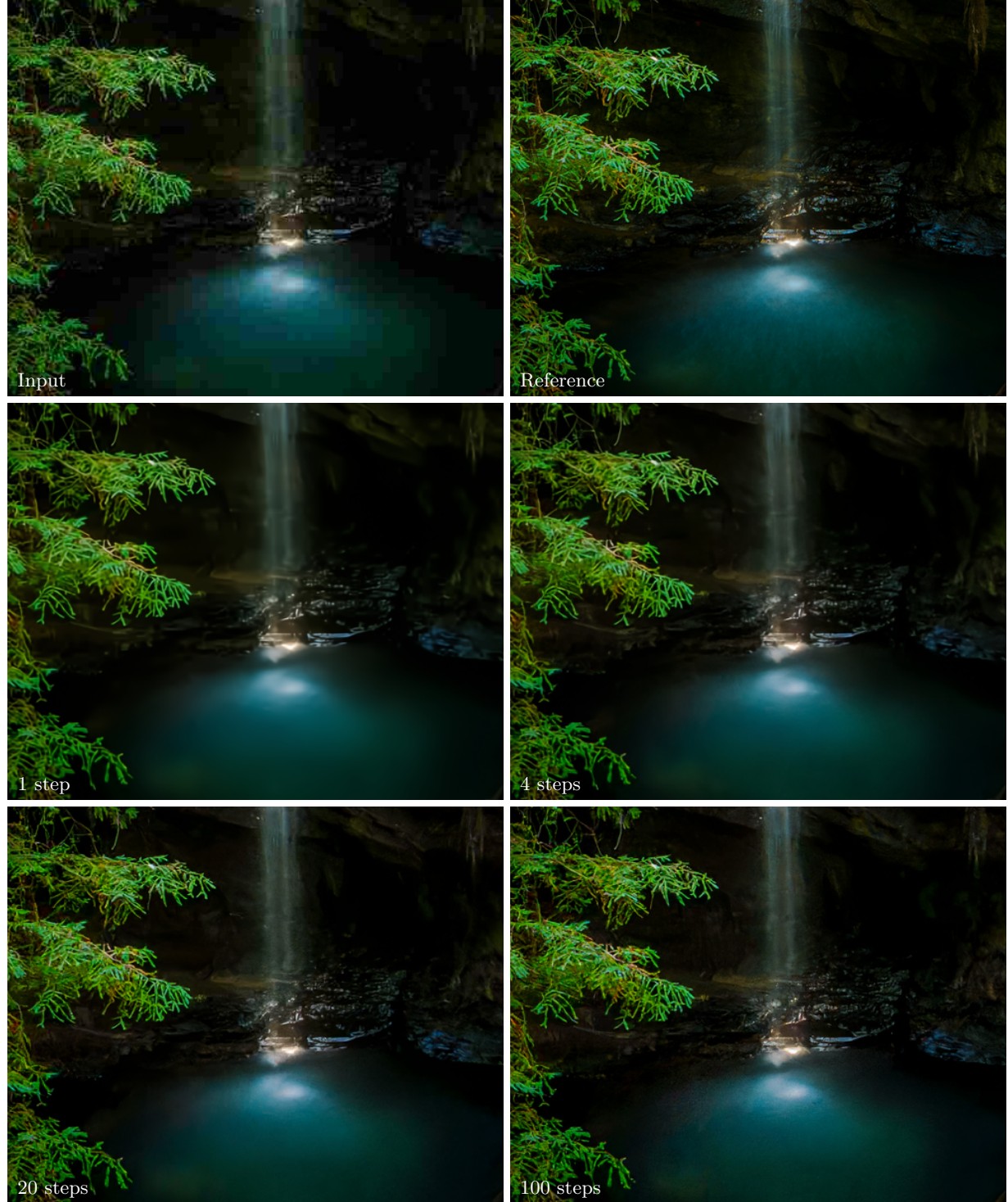

Figure 21: JPEG compression artifact removal results (quality factor 15, div2k test dataset). The proposed method (InDI) applied with different number of reconstruction steps. Best viewed electronically.

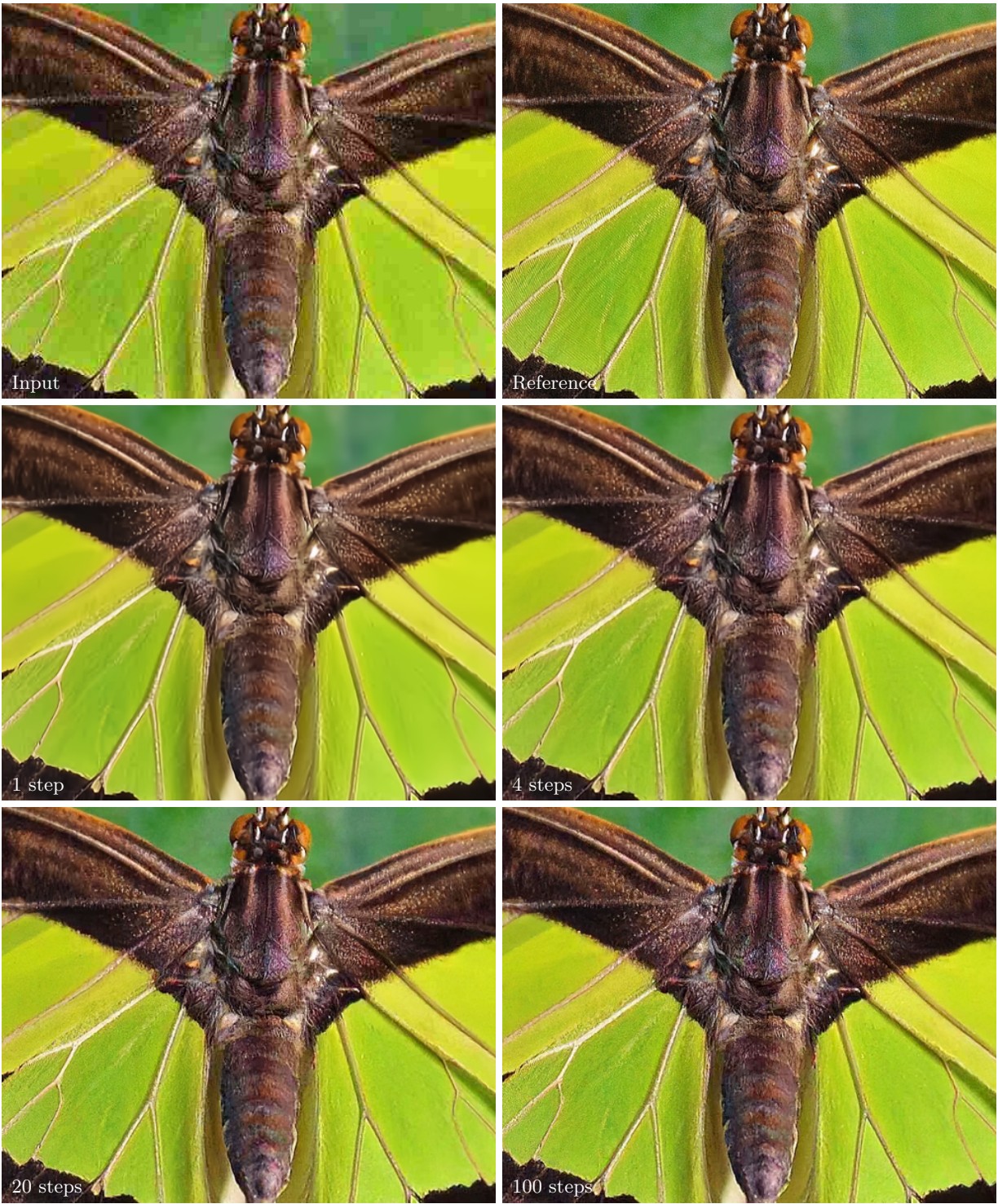

Figure 22: JPEG compression artifact removal results (quality factor 15, div2k test dataset). The proposed method (InDI) applied with different number of reconstruction steps. Best viewed electronically.

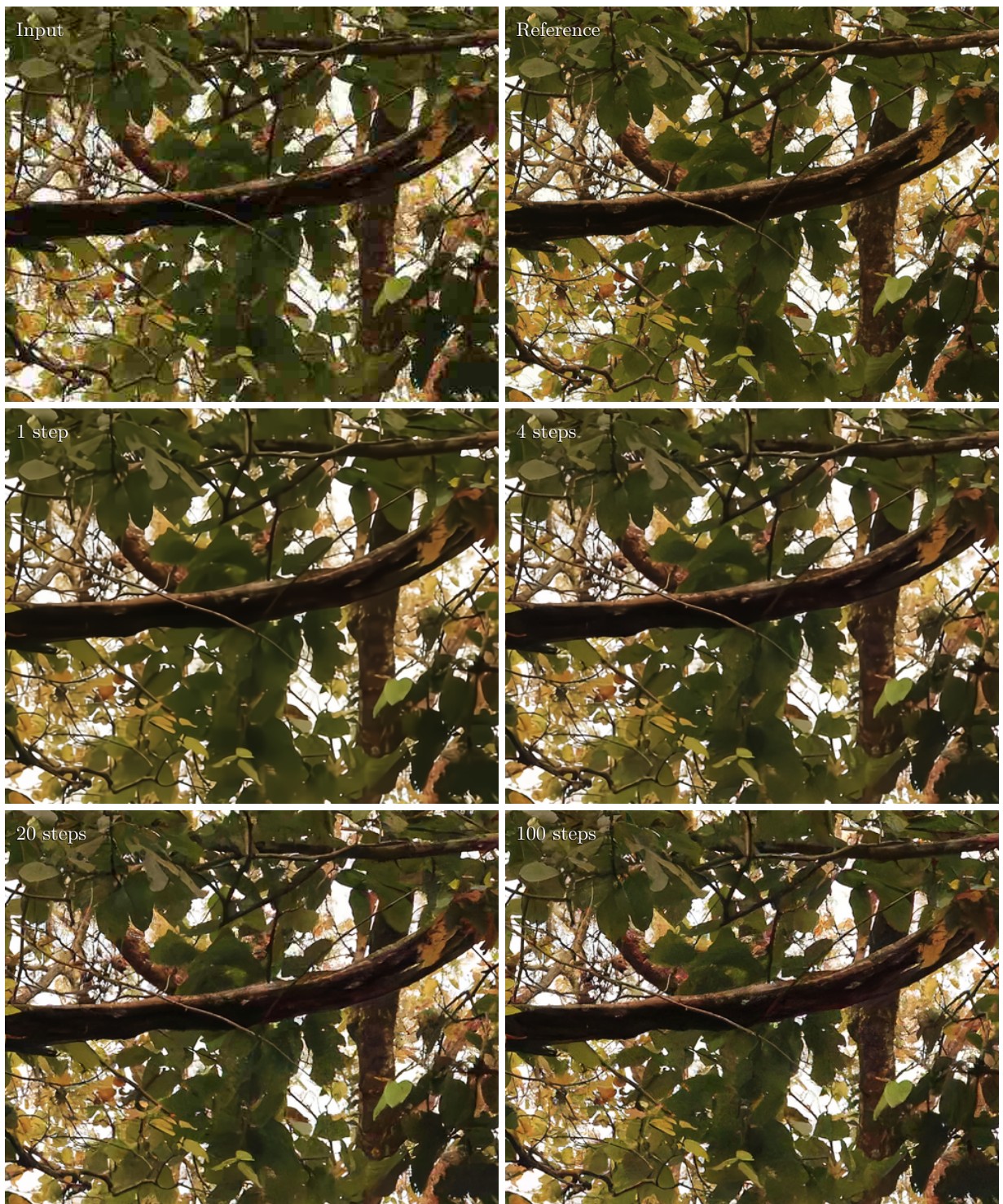

Figure 23: JPEG compression artifact removal results (quality factor 15, div2k test dataset). The proposed method (InDI) applied with different number of reconstruction steps. Best viewed electronically.

