# OpenReview forum: "Inversion by Direct Iteration: An Alternative to Denoising Diffusion for Image Restoration"
_TMLR — Accepted by TMLR_

### Review · Reviewer_9cRu · 2023-04-17

**Summary Of Contributions:**

The authors present a novel image restoration technique for the supervised learning setting which allows improved perceptual quality. The core of the method is to iteratively refine a solution in small steps. This allows to avoid the regression to the mean effect and get a solution closer to the image manifold.

**Audience:**

Yes

**Broader Impact Concerns:**

No concerns on broader impact.

**Claims And Evidence:**

Yes

**Requested Changes:**

I think the paper could be strengthened by some revisions and the inclusion of additional material.

I would like to see an extended set of comparisons with state-of-the-art methods where the extreme case of 1 iteration (which should maximize PSNR) is also shown. Additionally, it could be interesting to see results on the denoising problem. There are also questions about fairness of comparisons since the network architecture is different from other baselines. What would be the performance of the architecture when trained in a conventional manner?

It could also be interesting to discuss if the authors think there are connections between their work and the literature on plug and play methods (https://arxiv.org/abs/2203.17061).

As minor points: a few typos should be fixed throughout the paper; sections 2 and 3 are partially redundant and could be merged in a single section.

**Strengths And Weaknesses:**

Strenghts:
- The method is generally sound and an innovative technique to deal with the well-known distortion-perception tradeoff.
- Controlling the number of steps seems to achieve different perception-distortion tradeoff which might be useful in certain applications
- generally good experimental results

Weaknesses:
- as it is an iterative method, inference can be quite expensive
- performance does not always seem to be state of the art, neither in minimizing distortion nor in maximizing visual quality (see e.g. Fig.5)

---

> ### Author Response · Authors · 2023-05-11
> **Author Response**
>
> We thank the Reviewer for their valuable feedback. We are glad that the Reviewer found our approach sound and innovative, and well-validated through experimental results.
> Below, we address some questions/concerns raised by the Reviewer.
>
> **Comparison to SOTA, Other architectures.**
> As the reviewer points out, the core of the paper is a new idea on how we can actually generalize regression-based estimators into an iterative framework for trading distortion for perception quality. As such, it was not the main focus of this work to develop a state-of-the-art method for any particular application, or to analyze the many different powerful existing architectures (having different backbones, using attention mechanisms, transformers, and different types of normalizations). We see the reviewer’s very interesting suggestion as a great line of followup research that can adapt the proposed framework to become state-of-the-art for particular applications. Meanwhile, to motivate the present works usefulness, we included a number of comparisons to state-of-the-art models as reference, without the intention of showing an improvement over them.
>
> **Connection to Plug-n-play methods.**
> Thanks for the suggestion. We will incorporate a discussion about how these ideas can be connected to pnp/red type of formulations. We also believe that this could be an interesting line for future research.
>
> **Redundancy of Sections 2 and 3.**
> Thanks for pointing this out. We will compact both sections trying to avoid as much redundancy as possible. We will proofread the paper for typos.

---

### Review · Reviewer_6LtM · 2023-04-24

**Summary Of Contributions:**

This work presents a novel formulation for supervised image restoration, which aims to circumvent the so-called "regression to the mean" effect. The proposed algorithm functions by iteratively restoring low-quality input images through a series of small steps. The underlying intuition is that taking smaller restoration steps can largely avoid the regression-to-the-mean effect, as the set of plausible minor changes is relatively small.

The authors justify their approach by examining a simple 2D model, illustrating that the minimum mean-squared error (MMSE) optimal solution is susceptible to the "regression to the mean" effect. In contrast, iterative restoration converges to one of the plausible modes. The authors further support their claims by applying the method to various image restoration tasks, demonstrating that the Indi algorithm produces high-quality results for image denoising tasks.

Overall, this is an innovative and straightforward approach to image restoration, offering more strengths than weaknesses. Consequently, I recommend acceptance of the paper following minor revisions.



**Audience:**

Yes

**Broader Impact Concerns:**

No broader impact concerns are noted by the reviewer.

**Claims And Evidence:**

Yes

**Requested Changes:**

I recommend the following major changes (1, 2) to address the weaknesses identified in the paper:

1) Weakness-1: The authors should provide a mathematical argument, corresponding to the 2D toy problem, that demonstrates how INDI can alleviate the regression to the mean effect. This argument should be based on the closed-form solution for the 2D experiment and should explain whether INDI reduces the degeneracy of the operator H.

2) Weakness-2: To provide a more comprehensive evaluation, the authors should include image inpainting experiments in the experiments section of the paper.

Additionally, I suggest a minor change:

3) The authors should clarify the insights provided by the residual ODE flow and explain its relevance to the overall work.

**Strengths And Weaknesses:**

Strengths:

1) The described algorithm is remarkably straightforward to implement and does not necessitate any knowledge of the degradation process's analytic form, distinguishing it from generative denoising diffusion models.

2) The paper is well-written, with clear motivation and easy-to-understand proofs. The implementation process is presented in a user-friendly manner.

3) The authors effectively showcase the performance of their approach across various image restoration tasks, comparing it to both cold diffusion and conditional denoising diffusion models.

4) The authors also reveal the significant impact of bias during training time (with p(t) skewed towards t=1) on performance. This finding highlights the importance of the iterative procedure's increased certainty in determining the appropriate direction during the initial stages of the process.

Weakness:
1) Although the authors demonstrate the "regression to the mean" effect using a 2D toy model, they do not fully explain how INDI mitigates this effect. They claim that "the set of plausible 'slightly-less-bad' images is relatively small," but the reasoning behind this statement remains unclear from the 2D example. Consider the following example for the MMSE case. Given the model $y = Hx + n$ and the specified prior distribution $p(x) = \sum_{i=1}^d w_{i} \delta_{x-c_{i}}$, the MMSE solution is $\int x p(x|y) dx = c \int x p(y|x)p(x)dx$, with $c = \frac{1}{p(y)}$. The posterior distribution can be expressed as the product of the likelihood term and the prior. Under the Gaussian noise assumption, it becomes $c \int x \exp^{-\frac{-||y - Hx||_{2}^2}{\sigma^2}} p(x) dx$, where $p(x)$ is the discrete multimodal distribution.

Consequently, the integral simplifies to the sum over the four modalities: $\sum_{i=1}^4 w_{i} c_{i} \exp^{-\frac{-||y - Hc_{i}||_{2}^2}{\sigma^2}}$.

For the first denoising model, where $H = I$, the regression to mean effect is apparent, as it is an average over the $c_{i}$'s weighted by the distance of $y$ to each $c_{i}$. In the second inpainting problem, the non-trivial null-space of $H$ causes the y-coordinate information to be erased, and the MMSE iterate converges to a point where the x-axis coordinate matches that of the true modalities, while the y-axis of the MMSE solution is simply the average of the y-axis of the modalities. The "regression to the mean effect" is thus a joint result of the multimodal distribution of $p(x)$ and the non-trivial null-space of $H$ (and possibly the condition number in certain cases).

Considering this observation, how does INDI's update equation (equation 2) with the closed-form solution for the 2D model help alleviate the regression to mean effect? Is it because $E[x_{0}|\hat{x_{t}}]$ depends on $H_{t}$, which no longer has a null-space? A more detailed mathematical explanation of how INDI mitigates the regression to mean effect would be appreciated.

2) The ODE in equation-6 should be revised to reflect the residual ODE flow. However, the motivation for the ODE flow in this case remains unclear. It would be helpful if the authors could elaborate on what additional insights the ODE flow provides in this context.

3) It is surprising that the authors did not include any image inpainting experiments in their study. Given the current motivation and implementation of INDI, image inpainting appears to be a highly suitable task for the algorithm. The inclusion of such experiments would strengthen the paper and provide further evidence of INDI's applicability across different image restoration tasks.

---

> ### Author Response · Authors · 2023-05-12
> **Authors response**
>
> We thank the Reviewer for their valuable feedback. We are happy to see  that the Reviewer found the paper simple, clean, well motivated, and effectively validated in many image restoration tasks. Below, we address some concerns raised by the Reviewer.
>
> **Toy Example –  Inpainting.**
> The key observation, as the reviewer points out, is that $H_t$ is invertible for $t<1$ (i.e. trivial null-space). This implies that only on the very first step there is a difference between the general case and the basic denoising case. Namely, the first step of InDI in the inpainting case would move the point towards the posterior mean (given by the update rule after Eq (3), with $H_t = H$). The following steps will be applied with an $H_t$ that is invertible due to the regularization with a scaled identity matrix. This is observed in InDI's path (Figure 1b): the first step moves the point towards the $y=0$ axis, but the subsequent steps move the point towards the data manifold.
>
> We will introduce further discussion on this example to make this observation clear.
>
> **Inpainting – Experiments.**
> In the current manuscript we have illustrated InDI in five different tasks: motion deblurring, defocus deblurring, image super-resolution, jpeg artifact removal and generative modeling, including many ablations studies (noise level, inference algorithm, number of reconstruction steps). We agree that inpainting is an interesting problem. Yet, properly addressing inpainting requires a significant amount of additional work and time before we can submit the revision.
> We sincerely believe this additional effort would not bring much more insight to the approach than the example use cases already provided.
>
> **Residual ODE.**
> Thanks for pointing this out. First, we noted a typo in the Equation 6 shown in the submitted paper. Namely, the correct ODE equation is actually a residual flow:
> $$
> \frac{d\textbf{x}_t} {dt} = \frac{\textbf{x}_t - F(\textbf{x}_t,t)}{t}.
> $$
>
> We see at least two main uses for the residual ODE flow. First, it provides an analytical continuous formulation of the ideal update rule. This implies that we can adopt other discretizations, using standard ODE solvers, beyond the one that we derived. In particular, we could use ideas from recent work on speeding up diffusion models (see, e.g., [Consistency Models](https://arxiv.org/abs/2303.01469)). This line of follow-on work could lead to an improvement in inference speed. We will add a note in the revised manuscript to this effect.
>
> A second use of the residual ODE flow is for analyzing InDI's behavior on specific (e.g., simple) cases. For example, if we assume that the data prior $p(\textbf{x})$ is Gaussian, and we focus on denoising as the restoration task, then we can develop as follows (This analysis will be incorporated in an Appendix).
>
> **Denoising with a Gaussian Prior**
>
> Let's analyze InDI's behavior in the particular case where $p(\textbf{x}) = N(\textbf{c}, \sigma^2_c \textbf{I})$, and the restoration task is denoising $\textbf{y} = \textbf{x} + \textbf{n}$, where $\textbf{n}$ is white Gaussian of fixed standard deviation $\sigma_N$.  Then, $p(\textbf{y} | \textbf{x}) = N(\textbf{x}, \sigma^2_N \textbf{I})$ and $\mathbb{E}[\textbf{x}|\textbf{x}_t]= \frac{\sigma^2_c \textbf{x}_t + t^2\sigma^2_N \textbf{c}}{\sigma^2_c + t^2\sigma^2_N}$,
> where we have used the fact that $\textbf{x}_t = (1-t)\textbf{x} + t \textbf{y} = \textbf{x} + t\textbf{n}$.
>
> InDI's ideal ODE is given by
> $
> \frac{d \textbf{x}_t} {dt} = \frac{\textbf{x}_t-\mathbb{E}[\textbf{x} | \textbf{x}_t]}{t},
> $
> which in this case becomes:
> $$
> \frac{ d\textbf{x}_t}{d t} = \frac{t \sigma^2_N (\textbf{x}_t-\textbf{c})}{\sigma^2_N t^2 + \sigma^2_c}.
> $$
> We are interested in solving this equation at $t=0$, with boundary condition $\textbf{x}_1=\textbf{y}$ at $t=1$.  This is a separable ODE having general solution:
> $\textbf{x}_t = \textbf{c} + (\textbf{y} - \textbf{c}) \sqrt{ \frac{t^2 + \alpha^2}{1 + \alpha^2}}$,
> where $\alpha = \frac{\sigma_c}{\sigma_N}$. The solution at $t=0$, is
> $$
> \textbf{x}_\text{InDI} = \textbf{x}_0 = \textbf{c} + (\textbf{y}-\textbf{c}) \sqrt{\frac{\sigma^2_c}{\sigma^2_c + \sigma^2_N}}.
> $$
>
> Note that $\mathbb{E}[\textbf{x}_\text{InDI}] = \textbf{c} = \mathbb{E}[\textbf{x}]$,
> and the covariance $\text{var}(\textbf{x}_\text{InDI}) = \text{var}(\textbf{y}) \frac{\sigma^2_c}{\sigma^2_c + \sigma^2_N} = \text{var}(\textbf{x})$,
> since $\text{var}(\textbf{y}) = (\sigma^2_c + \sigma^2_N)\textbf{I}.$ Given $p(\textbf{x})$ and $p(\textbf{x}_\text{InDI})$ are Gaussian distributions with same mean and covariance, we have $p(\textbf{x}_\text{InDI})=p (\textbf{x})$. That is, in the limit, InDI  generates samples from the prior distribution $p(\textbf{x})$.
>
> It is worth noting that in this case the MMSE and MAP estimates coincide,
> $$
> \textbf{x}_\text{MMSE}  = \textbf{x}_\text{MAP} = \frac{\sigma^2_c \textbf{y} + \sigma^2_N \textbf{c}}{\sigma^2_c + \sigma^2_N},
> $$
> and are in fact different from InDI's estimate.

---

### Review · Reviewer_16FV · 2023-05-05

**Summary Of Contributions:**

This paper proposes a novel diffusion-based image restoration process (Inversion by Direct Inversion) designed to solve imaging inverse problems without explicit knowledge of the degradation forward process. The proposed technique assumes access only to paired clean and degrades images (x,y)~p(x,y); it does not need to know p(y|x). That is, unlike other diffusion techniques (e.g., cold diffusion) it does not need to know the degradation process. The proposed method then constructs a series of intermediates samples x_{t_1}, x_{t_2}, ... from convex combinations of x and y. It trains a restoration model to reconstruct x from each of these samples, effectively training a network to compute E[x|x_t]. This restoration model is repeatedly applied (with a principled weighting scheme) to reconstruct x. If the restoration model uses only a few large steps it has little distortion (high PSNR) while if it is applied many iterations with small steps it has higher perceptual quality. The authors apply the proposed method to motion deblurring, super-resolution, defocus deblurring, and JPEG artifact removal. It offers SOTA performance at motion deblurring and is competitive with existing techniques on the other tasks.


**Audience:**

Yes

**Broader Impact Concerns:**

I have no concerns about this work. Like other generative techniques, it could potentially be misused by bad actors.

**Claims And Evidence:**

Yes

**Requested Changes:**

I support accepting the paper, but addressing the following would strengthen the work:

Overall, the paper provides a comprehensive survey of the literature and related work. The recent papers [A] and the follow-up [B] (concurrent) seem closely related and may be worth mentioning.

[A] Luo, Ziwei, et al. "Image Restoration with Mean-Reverting Stochastic Differential Equations." arXiv preprint arXiv:2301.11699 (2023).

[B] Luo, Ziwei, et al. "Refusion: Enabling Large-Size Realistic Image Restoration with Latent-Space Diffusion Models." arXiv preprint arXiv:2304.08291 (2023).

Highlighting the best and second best results in all tables (as they were done in table 1) would make them easier to read and interpret.


#### Typos and suggestions

Pg 10: "Table 5(a)"-->"Figure 5(a)".

Pg 10: I suggest making 5(b) Table 2

Pg 10: It's not obvious whether the "ours" in 5(b) had noise injection or not. The adjacent figure 5(a) suggests this is an important distinction to make.

**Strengths And Weaknesses:**

Strengths:

-Well-motivated

-Very clearly written

-Interesting

-Thorough ablation studies


Weaknesses:

-A few minor presentation issues

-Not state-of-the-art at everything

-Doesn't generalize (requires retraining per inverse problem)

---

> ### Author Response · Authors · 2023-05-11
> **Author Response**
>
> We thank the Reviewer for their valuable feedback. We are glad that the Reviewer found the paper well-motivated, clean and interesting.
>
> **Missing references.** Thanks a lot for pointing out these related works. We will incorporate them with other relevant references that we collected below.
>
> 1. Welker, Simon, Henry N. Chapman, and Timo Gerkmann. 2022. Blind Drifting: Diffusion models with a linear SDE drift term for blind image restoration tasks. NeurIPS  Workshop, The Symbiosis of Deep Learning and Differential Equations II.
> 2. Luo, Z., Gustafsson, F.K., Zhao, Z., Sjölund, J. and Schön, T.B., 2023. Image Restoration with Mean-Reverting Stochastic Differential Equations. arXiv preprint arXiv:2301.11699
> 3. Liu, G.H., Vahdat, A., Huang, D.A., Theodorou, E.A., Nie, W. and Anandkumar, A., 2023. I^2 SB: Image-to-Image Schrödinger Bridge. arXiv preprint arXiv:2302.05872.
> 4. Song, Y., Dhariwal, P., Chen, M. and Sutskever, I., 2023. Consistency models. arXiv preprint arXiv:2303.01469.
> 5. Albergo, M.S., Boffi, N.M. and Vanden-Eijnden, E., 2023. Stochastic interpolants: A unifying framework for flows and diffusions. arXiv preprint arXiv:2303.08797.
> 6. Luo, Z., Gustafsson, F.K., Zhao, Z., Sjölund, J. and Schön, T.B., 2023. Refusion: Enabling Large-Size Realistic Image Restoration with 7. Latent-Space Diffusion Models. arXiv preprint arXiv:2304.08291.
>
> **Typos and suggestions.** We will incorporate all the suggestions and proofread the updated manuscript for typos.

---

### Decision · Action_Editors · 2023-06-13

**Recommendation:** Accept as is

**Comment:**

This is a very well-written paper on a novel formulation for supervised image restoration. All reviewers appreciate the simple yet effective framework and solid evaluation on five different tasks (motion deblurring, super-resolution, defocus deblurring, jpeg artifact removal, and generative modeling), demonstrating the proposed method's applicability to various image restoration problems.

The notable strengths by the reviewers are:

"remarkably straightforward to implement" - Reviewer 6LtM
"clear motivation and easy-to-understand proofs" - Reviewer 6LtM
"Thorough ablation studies" - Reviewer 16FV
"The method is generally sound and an innovative technique to deal with the well-known distortion-perception tradeoff" - Reviewer 9cRu

There were some missing references and discussions of connection with prior work raised by the reviewers. After the authors' responses, all reviewers strongly support accepting the paper. The AE appreciates the simplicity of the work and the general applicability of the proposed method on different problems. While, the current method does not outperform the state-of-the-art on each individual task, the AE agrees that this is not the main focus of the paper. The method's simplicity and promising performance may inspire many follow-up research. The AE thus recommends "accepting as is" with featured certification.




**Audience:**

Yes, the paper will likely be of interests for a TMLR's audience.

**Claims And Evidence:**

Yes, the experiments have sufficiently supported the claims of the paper. While one reviewer requests adding image inpainting experiments,  the AE agrees that it's not required due the computational cost and the marginal evidence beyond the five applications already included in the paper.